# NRF2 supports non-small cell lung cancer growth independently of CBP/p300-enhanced glutathione synthesis

Ryan J Conrad[1,8], James A Mondo[2,8], Mike Lingjue Wang [ID][3], Peter S Liu[4], Zijuan Lai[3], Feroza K Choudhury[3], Qingling Li[4], Weng Ruh Wong[4], James Lee[1], Frances Shanahan[1], Eva Lin[1], Scott Martin[1], Joachim Rudolph [ID][5], John G Moffat[6], Dewakar Sangaraju[3], Wendy Sandoval[4], Timothy Sterne-Weiler [ID][1,7 ✉] & Scott A Foster [ID][1 ✉]

## Abstract

**Nuclear factor erythroid 2-related factor 2 (NRF2) is a stress responsive transcription factor that is mutationally activated in a subset (~25%) of clinically-aggressive non-small cell lung cancers (NSCLC). Mechanistic insight into drivers of the NRF2 dependency remains poorly understood. Here, we defined a novel NRF2 target gene set linked to NRF2-dependency in cancer cell lines, and observed that a significant portion of these genes is devoid of promoter-proximal NRF2 occupancy. Using integrated genomic analyses, we characterized extensive NRF2-dependent enhancer RNA (eRNA) synthesis and NRF2-mediated H3K27ac deposition at proximal and distal enhancer regions regulating these genes. While CBP/p300 is a well-validated direct interaction partner of NRF2 with prominent functions at enhancers, we report that this inter- action is not required for NRF2-dependent NSCLC cell growth, indicating that NRF2 can sustain sufficient transcriptional activity in the absence of CBP/p300 coactivation. Broad metabolic profiling established a primary role for CBP/p300 in NRF2-dependent accumulation of glutathione and glutathione-related metabolites. While redox homeostasis via enhanced glutathione production is commonly associated with the normal physiological role of NRF2, collectively our results suggest that NRF2-dependent cancer cell growth does not require this enhanced glutathione production.**

**Keywords** NRF2; CBP/p300; Non-small Cell Lung Cancer; Glutathione; ROS
**Subject Categories** Cancer; Chromatin, Transcription & Genomics; Metabolism

## Introduction

Transcriptional dysregulation is a common feature of human cancers (Bradner et al, 2017). Aberrant activation of specific transcriptional programs has the capacity to greatly alter cellular state, driving oncogenesis and supporting cancer cell proliferation. The nuclear factor erythroid 2-related factor 2 (*NFE2L2* or NRF2) transcription factor is constitutively activated in many cancers and promotes a specific gene expression program involved in cellular detoxification, redox buffering, and metabolic rewiring (Cancer Genome Atlas Research, 2012; Cancer Genome Atlas Research et al, 2017; DeNicola et al, 2011; Rojo de la Vega et al, 2018). Under homeostatic conditions, transient activation of NRF2 is cytoprotective, allowing for adaptation to various cell intrinsic (i.e., reactive oxygen species (ROS) generated by mitochondrial function) and extrinsic (i.e., xenobiotic) stressors (Kensler et al, 2007; Ma, 2013). Constitutive activation of NRF2, mainly via loss-of-function mutations in the CUL3-adapter *KEAP1* or gain-of-function mutations in *NFE2L2*, is thought to provide a fitness advantage to tumor cells in confronting metabolic/oxidative stress resulting from dysregulated growth (DeNicola et al, 2011; Rojo de la Vega et al, 2018). In patients, activation of the NRF2 pathway is associated with more aggressive disease (Homma et al, 2009; Solis et al, 2010) and decreased response to various classes of cancer therapeutics (Gadgeel et al, 2020; Jaenne et al, 2022; Skoulidis et al, 2021).

The NRF2 target gene set ranges from ~30–500 genes depending on publication (Kim et al, 2016; Malhotra et al, 2010; Mitsuishi et al, 2012; Romero et al, 2017; Singh et al, 2021), Genes common among published target sets include those related to redox biology (i.e., *GCLC*, *SLC7A11*, *TXN*) and the pentose-phosphate pathway (PPP, i.e., *PGD*, *TALDO1*), in addition to many NADP-dependent metabolic enzymes (i.e., *AKR1C1*, *AKR1B10*, and *CYP4F11*). However, the relative contribution of NRF2 target genes to NRF2-dependent cancer cell proliferation is currently unknown. Furthermore, functional inter- rogation of the corresponding importance of pathways downstream of NRF2-dependent transcription in cancer remains limited.

[1]Department of Discovery Oncology, Genentech, Inc., South San Francisco, CA 94080, USA. [2]Roche Informatics, F. Hoffmann-La Roche Ltd., Mississauga, ON, Canada. [3]Department of Drug Metabolism and Pharmacokinetics, Genentech, Inc., South San Francisco, CA 94080, USA. [4]Department of Microchemistry, Proteomics & Lipidomics, Genentech, Inc., South San Francisco, CA 94080, USA. [5]Department of Discovery Chemistry, Genentech, Inc., South San Francisco, CA 94080, USA. [6]Department of Biochemical and Cellular Pharmacology, Genentech, Inc., South San Francisco, CA 94080, USA. [7]Department of Oncology Bioinformatics, Genentech, Inc., South San Francisco, CA 94080, USA. [8]These authors contributed equally: Ryan J Conrad, James A Mondo. ✉E-mail: sternewt@gene.com; foster.scott@gene.com

NRF2 is a member of the cap 'n' collar (CNC)-bZip transcription factor family that upon recognition of the antioxidant response *cis* element (ARE) heterodimerizes with small musculoa-poneurotic fibrosarcoma (sMAF) proteins to activate transcription (Igarashi et al, 1994; Itoh et al, 1997; Moi et al, 1994). NRF2 also cooperates with components of the transcriptional machinery (i.e., MED16) (Sekine et al, 2016), chromatin remodelers (i.e., BRG1) (Zhang et al, 2006) and chromatin-modifying enzymes (i.e., CBP) (Katoh et al, 2001) to drive target gene expression. The CBP lysine acetyltransferase is a well-recognized, direct NRF2 binding partner (Tonelli et al, 2018), contributing to NRF2 transcriptional activity at reporter genes (Katoh et al, 2001) and responsible for acetylating NRF2 itself (Sun et al, 2009). The CBP paralog p300 has also been linked to NRF2 function (Sekine et al, 2016). CBP/p300 is thought to act as a transcriptional coactivator through deposition of H3K27ac marks (Creyghton et al, 2010; Tie et al, 2009), yet the significance of CBP/p300 (and H3K27ac) to NRF2 activity has not been explored for the broader NRF2 transcriptome.

The prevailing model for NRF2-dependent target gene expression involves the binding of NRF2 to ARE sequences within proximal promoters of specific target genes (Tonelli et al, 2018). However, accumulating ChIP-seq data indicates that NRF2 predominantly binds intronic or distal intergenic regions (Malhotra et al, 2010; Namani et al, 2019). A cooperative role of NRF2 and C/EBPβ (a transcription factor of the CCAAT/enhancer binding protein family) was described in generating tumor-specific enhancers that promote tumor initiation via *NOTCH3* activation (Okazaki et al, 2020). However, the extent to which NRF2 regulates its target genes beyond promoter-restricted regions, and the significance of this regulation to NRF2-dependent cancer cell maintenance, remains unexplored.

Here we defined a signature of 59 genes and two metabolites (GSH and NADP) that is robustly associated with NRF2 dependency across an extensive panel of cancer cell lines. In analyses of ChIP-seq data, we observed that many of these genes were devoid of NRF2 in their promoter regions. Integrating ChIP-seq, HiC, and PRO-seq datasets, we revealed broad functional binding of NRF2 at enhancer elements, including NRF2-dependent enhancer-derived RNA (eRNA) synthesis and extensive linkages between NRF2 target promoters and enhancers. Mechanistically, we found that NRF2 is required for the deposition of H3K27ac marks at sites of nascent transcription, including eRNAs. Disabling CBP/p300-binding to NRF2 broadly attenuated NRF2 transcriptional activity resulting in an altered metabolic (depletion of glutathione) and lipidomic (specific depletion of triglycerides) cellular landscape. Surprisingly, these phenotypic consequences of CBP/p300 inhibition did not result in viability defects in NRF2-dependent cell lines. Collectively, these observations establish that NRF2 alone can sustain sufficient transcriptional output to maintain cancer cell line growth and that metabolic reprogramming of cells driven by NRF2 does not require redox buffering via enhanced glutathione production.

# Results

## Absence of promoter proximal NRF2 at NRF2 signature genes

To identify critical pathways downstream of NRF2 activation in cancer, we sought to define a core set of genes and other features associated with NRF2-dependent growth across an expansive panel of

cancer cell lines. We approached this by identifying omics- features (i.e., expression, mutation, copy number, pathways, metabolites, etc.) that are predictive of NRF2 CRISPR knockout scores from the DepMap. Using random forest regression as the predictive model, we identified pathways commonly associated with NRF2, such as reactive oxygen species (ROS) and xenobiotic metabolism, in addition to reduced glutathione and NADP levels, as top features associated with NRF2 dependency (Fig. 1A and Table EV1; $R^2 = 0.437$, 95.2% mean AUC; see Methods). We next optimized a NRF2 transcriptional signature resulting in a set of 59 genes based on their contribution to the model and overlap with previously reported NRF2 signatures (Figs. 1A and EV1A,B, Methods). To evaluate NRF2-dependent regulation of these genes, we generated two *KEAP1*-mutant NSCLC cell lines (A549 and NCI-H460) stably integrated with a doxycycline (dox)-inducible shRNA targeting NRF2. We observed knockdown of NRF2 protein and robust growth inhibition upon dox treatment, highlighting the dependency of these cell lines on NRF2 for growth (Fig. EV1C,D). RNA-seq of dox-treated cells demonstrated strong suppression of the majority of these NRF2 signature genes in both cell lines at 6 and 24 h (Fig. EV1E).

To interrogate the mechanisms underlying NRF2-directed regulation of this core gene set, we compared mean promoter (1 kb restricted) ChIP-seq signal in A549 for NRF2, p300, as well as the histone marks H3K4me3 and H3K27ac, which are enriched at promoters and enhancers, respectively. We observed that many signature genes completely lack, or have very little, NRF2 ChIP-seq signal within 1 kb of their promoter (Fig. 1B). We defined two sets of genes, those either largely devoid of promoter bound NRF2 (Group I, green) or genes with high promoter bound NRF2 (Group II, purple) (Fig. EV1F). As defined by our analysis, we found that the majority of genes in both sets exhibited high levels of promoter-associated H3K4me3 and low levels of enhancer-associated H3K27ac. Despite the known direct association of p300 with NRF2, p300 is only detected at a small subset of group II gene promoters despite high occupancy of NRF2. Notably, NRF2 promoter binding was not determinant of transcriptional response to NRF2 knockdown in A549 and NCI-H460, which was highly correlated between the two models (Fig. 1C).

The observation that low promoter occupancy genes were highly responsive to NRF2 knockdown (Fig. 1C) led us to investigate whether NRF2 may be acting at proximal enhancer elements (>1 kb from the promoter region). Indeed, the nearest significant NRF2 ChIP peak (IDR q-value < 0.05) is ~15 kb away on average for group I genes, while it overlaps the majority of group II genes (Fig. 1D). In an unbiased interrogation of NRF2 function at enhancers, we intersected NRF2 peaks adjacent to target gene promoters (irrespective of distance) with histone and transcription factor ChIP datasets in A549 from ENCODE (Dataset EV1). This revealed an expected enrichment of promoter marks (in particular H3K4me3) in the group II gene set. However, for group I genes we observed a significant enrichment of enhancer-related histone marks (H3K27ac, H3K4me1), signifying *trans*-regulatory functions at these loci (Fig. 1E). This is further supported by the significant enrichment of p300 and RAD21, a member of the cohesin complex which can regulate long-range transcriptional activation via chromatin looping (Waldman, 2020) (Fig. 1F). Other notably enriched transcription factors include AP-1 and C/EBPβ, the latter of which has been previously linked to NRF2 activity at specific enhancers (Okazaki et al, 2020). These data provide strong evidence of pervasive NRF2 binding at proximal enhancers.

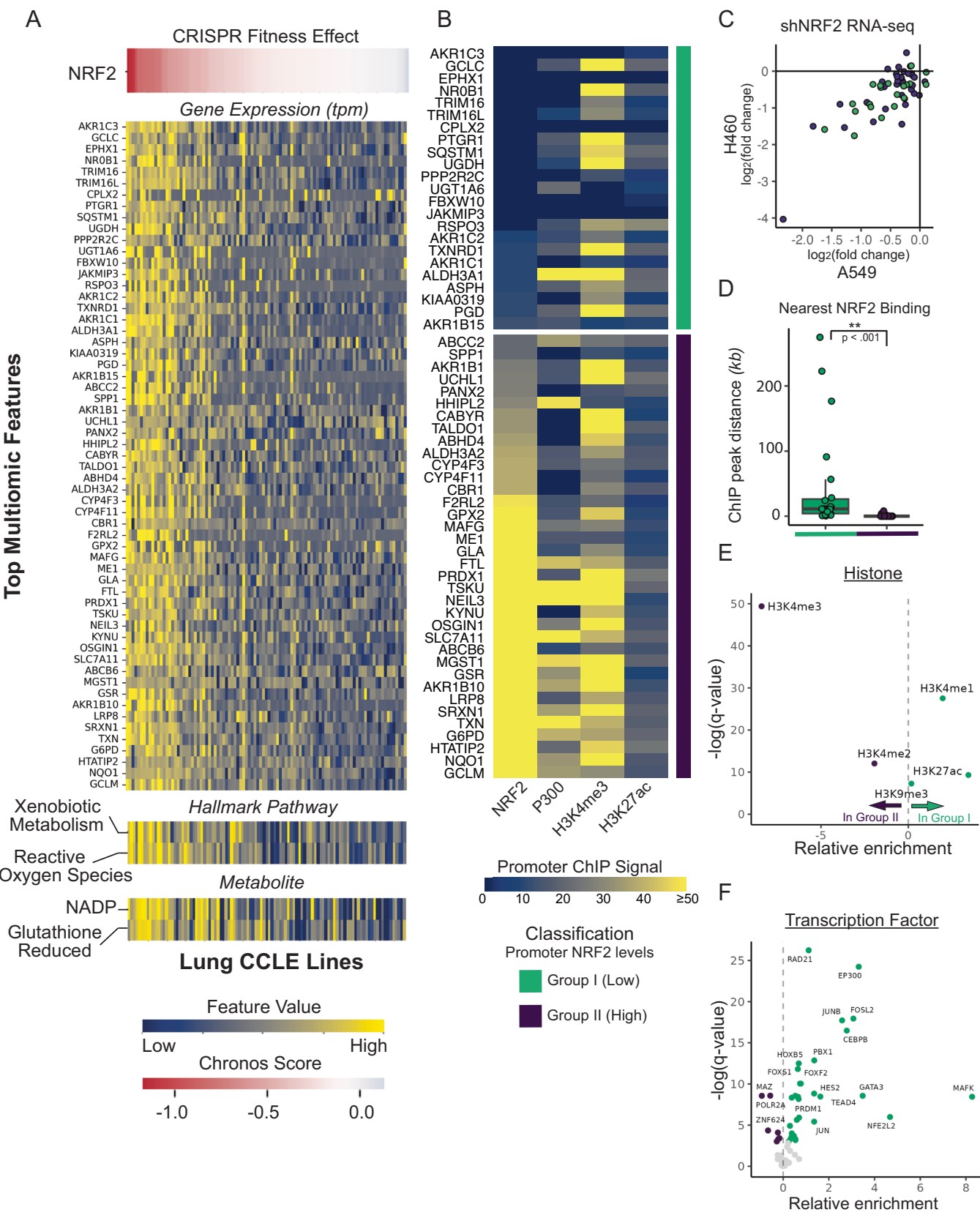

**Figure 1. Absence of promoter proximal NRF2 at highly responsive NRF2 target genes.**

(A) Top multi-omic features linked to NRF2 dependency in CCLE cell lines as defined by NRF2 DepMap Chronos score. Chronos scores estimate the fitness effect of a CRISPR gene knockout in a pooled whole genome screen, with a score of zero indicating no fitness effect and lower scores representing a decrease in cellular fitness. (B) Heatmap of promoter ChIP-seq levels for NRF2 signature genes for low promoter bound NRF2 (green; group I) and high promoter bound NRF2 (purple; group II) classes. Data is mined from publicly available ChIP-seq datasets in A549 cells (Dataset EV1). Signal represents the average of all peaks within 1 kb of the promoter. (C) Correlation between RNA-seq $\log_2$ fold change values upon NRF2 knockdown in A549 and NCI-H460-shNRF2 cells (500 ng/mL, 6 h) vs control (water) across group I and group II NRF2 signature genes. (D) Distance in kilobases from each gene's promoter to the nearest significant NRF2 ChIP peak. The box represents the first and third quartiles, with the median shown as a horizontal line. Whiskers extend to the smallest and largest value within 1.5 * the interquartile range. Group I, $n = 23$, group II, $n = 36$. P-values derived from a Wilcoxon rank-sum test, $p = 2.659e$-11. (E) Relative enrichment of histone ChIP-seq peaks overlapping significant NRF2 peaks proximal to NRF2 signature genes. These proximal NRF2 ChIP peaks are defined as a peak whose closest gene is a NRF2 signature gene, while not being within the gene region. Histone ChIP-seq data are mined from publicly available ChIP-seq datasets in A549 cells (Dataset EV1). The box represents the first and third quartiles, with the median shown as a horizontal line. Whiskers extend to the smallest and largest value within 1.5 * the interquartile range. (F) Relative enrichment of transcription factor ChIP-seq peaks overlapping significant NRF2 peaks proximal to NRF2 signature genes in A549 cells. Transcription factor ChIP-seq data are mined from publicly available ChIP-seq datasets in A549 cells (Dataset EV1). Source data are available online for this figure.

## Broad requirement of NRF2 for eRNA synthesis and H3K27 acetylation at enhancers

To characterize the role of NRF2 in directly orchestrating transcriptional activation via enhancer elements, we performed precision nuclear run-on sequencing (PRO-seq, (Mahat et al, 2016)) to quantify changes in nascent RNA following NRF2 knockdown in our A549 and H460 models (Fig. EV1C–E). We first summed all PRO-seq reads within annotated gene regions, identifying NRF2 signature genes among the most significant changes (Fig. EV2A, A549: 26 of the top 50 genes, H460: 26 of the top 50 genes).

Gene-agnostic sites of NRF2-dependent nascent transcription were next identified from bi-directional peaks significantly depleted in response to NRF2 knockdown in A549. Similar to our gene-centric analysis, promoter nascent RNA peaks of NRF2 signature genes were significantly impacted by shNRF2 treatment (38 of the top 50 peaks) (Figs. 2A and EV2B, purple). Sites were subsequently annotated by their relation to a proximal enhancer, as defined by ChromHMM (Figs. 2A and EV2B, dark orange). Notably, several of the top hits were sites of nascent transcription at enhancer elements (eRNAs) proximal to NRF2 target genes (i.e., AKR1C1 and ALDH3A1), establishing the requirement of NRF2 for eRNA synthesis at these loci. We extended these observations to NCI-H460, and found a similar enrichment of NRF2 signature genes at peaks responsive to NRF2 treatment (Fig. EV2C, 16 of the top 50 peaks), with highly responsive peaks at both promoters and proximal enhancers.

We next sought to identify interactions in higher-order chromatin architecture between distal enhancer elements and NRF2 signature genes. To achieve this, we simulated H3K27ac HiChIP using HiC and ChIP-seq data in A549 cells and observed links between NRF2 signature genes and distal enhancer elements responsive to NRF2 knockdown (Figs. 2A and EV2B, light orange). Comparing the classification of peaks associated with NRF2 signature genes relative to other responsive regions, we observed an increase in the proportion of higher-order linkage involving two or more enhancers (41% vs 17%) for NRF2 signature genes (Fig. 2B). Applying group I and II classifications of signature genes to this set of linked NRF2 responsive PRO-seq peaks, we observed significantly increased levels of NRF2 and p300 at distal enhancers linked to group I genes (Fig. 2C). Collectively, these findings define extensive, NRF2-dependent activity and p300

association across proximal and distal enhancer elements linked to NRF2 signature genes.

To address the mechanism through which NRF2 coordinates enhancer activity, we analyzed H3K27ac ChIP data from A549 cells treated with siRNA targeting NRF2 (Okazaki et al, 2020). We observed an overall reduction of H3K27ac levels following NRF2 knockdown, with many regions linked to NRF2 signature genes significantly impacted (Fig. 2D). To generalize these findings across additional cell lines, we next quantified H3K27ac ChIP for NRF2 responsive PRO-seq peaks across a panel of NRF2-dependent and -independent lung cell lines (Fig. EV2D, Table EV2). Comparing 26 NRF2-independent and 8 NRF2-dependent lines we found a significant increase in H3K27ac levels at promoter and proximal enhancer regions of NRF2 PRO-seq responsive sites (Fig. 2E). Interestingly, we did not observe a difference in sites identified as distal-linked enhancers in A549 across this panel of lines, in agreement with reported cell-line specificity of distal linkages (preprint: (Gschwind et al, 2023)). These data establish functional cooperation between NRF2 and CBP/p300, as well as place NRF2 upstream, in the deposition of H3K27ac marks at signature gene enhancers.

Genome browser tracks exemplify the gene-specific regulatory landscapes of NRF2 signature genes. For example, the group I gene *GCLC*, which encodes the catalytic subunit of the glutamate-cysteine ligase (GCL) necessary for the rate-limiting step of GSH biosynthesis, has low NRF2 promoter occupancy and high NRF2 binding at a proximal enhancer, the latter of which is linked to a distal enhancer (Fig. 2F). In contrast, *GCLM*, a group II gene encoding the regulatory subunit of GCL, has strong NRF2 promoter occupancy (Fig. 1B) with no proximal or distal links (Fig. EV3A). *AKR1C1*, *AKR1C2*, and *AKR1C3* are group I genes, which all have low promoter NRF2 (Fig. 1B), and appear to be driven by a large cluster of highly linked enhancers (Fig. EV3B, (Scalera et al, 2021)). Genome browser snapshots of all 59 NRF2 signature genes are included as an Appendix PDF.

## CBP/p300 is required for robust expression of NRF2 target genes

The requirement of NRF2 for eRNA synthesis and H3K27ac levels at enhancer regions across an array of NRF2-dependent cell lines establishes a widespread role for NRF2 in enhancer activity. Given CBP/p300 mediates acetylation of H3K27 genome-wide (Tie et al,

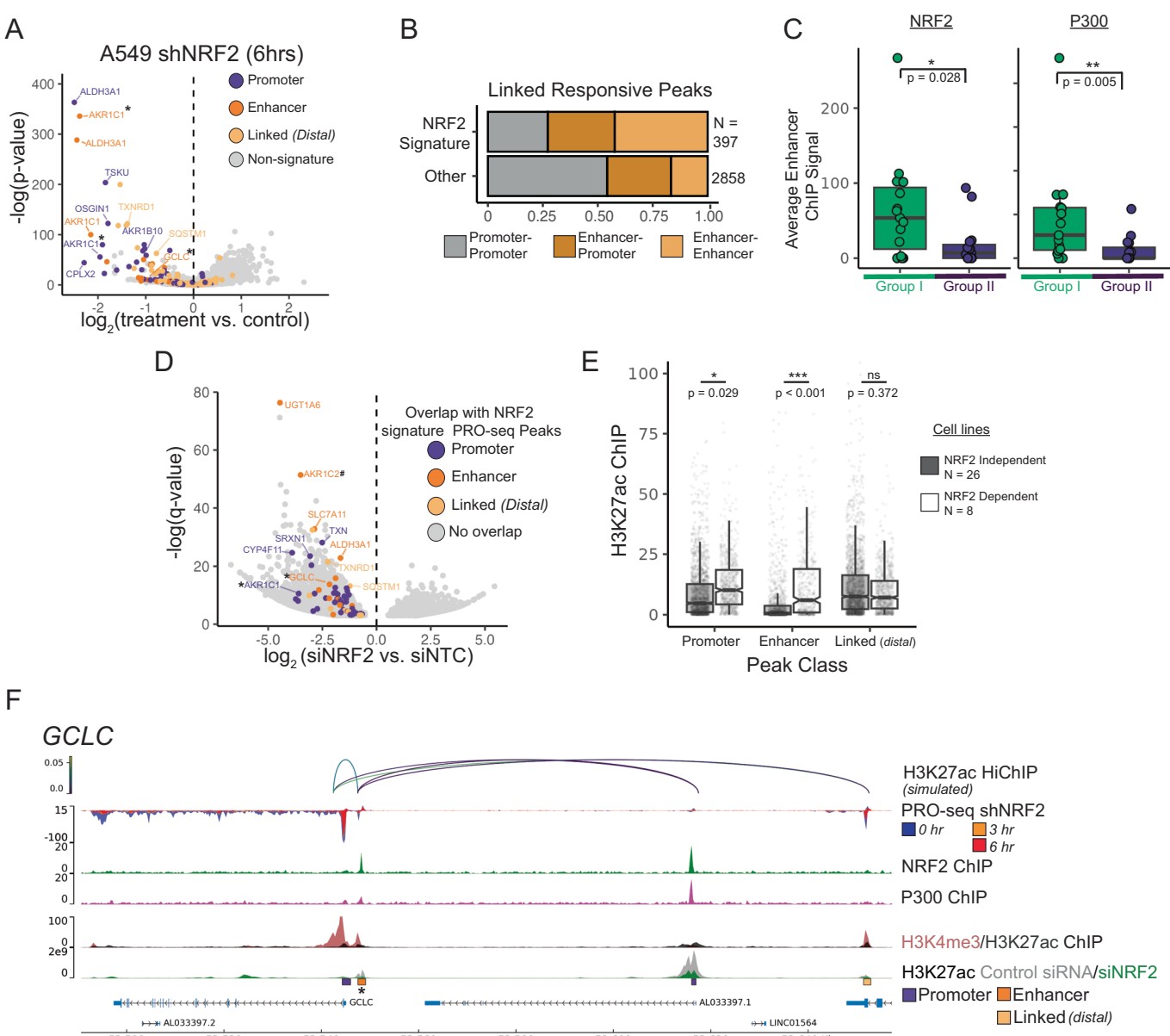

**Figure 2. Broad requirement of NRF2 for eRNA synthesis and H3K27 acetylation at enhancers.**

(A) Volcano plot of PRO-seq peaks upon NRF2 knockdown. Each point represents an individual peak. Peaks are colored by their relationship to NRF2 signature genes, either overlap with the promoter (purple), overlap with a ChromHMM annotated enhancer (dark orange) or overlap with a predicted linked distal enhancer (light orange). A549 shNRF2 treatment is 250 ng/mL dox at 6 h relative to control (water), and data represent three replicates. (B) Distribution of linkage types between shNRF2 responsive PRO-seq peaks in NRF2 signature vs. non-signature genes. (C) Average NRF2 (left) and p300 (right) ChIP levels across all proximal and linked enhancers identified from PRO-seq. Genes for which no linked enhancers were found are assigned a value of 0. The box represents the first and third quartiles, with the median shown as a horizontal line. Whiskers extend to the smallest and largest value within 1.5 * the interquartile range. P-values derived from a Wilcoxon rank-sum test. Group I $n = 23$, group II $n = 36$. (D) Log$_2$ fold change of H3K27ac signal between NRF2 siRNA vs non-targeting control (siNTC) at 48 h in A549 cells. Each point represents a ChIP-seq peak with significantly different levels of H3K27ac following NRF2 knockdown (FDR <0.05). These regions are then intersected with responsive PRO-seq peaks linked to NRF2 signature genes (Fig. 2A) and colored by linkage type. H3K27ac ChIP data generated by Okazaki et al, 2020. Data derived from three independent samples per condition (Dataset: GSE118840). (E) Average H3K27ac ChIP-seq signal (fold-change over input control) in NRF2-independent (gray) vs -dependent (blue) lines (Table EV2) within genomic regions overlapping with NRF2 responsive PRO-seq peaks. NRF2-independent cell lines are defined by a NRF2 Chronos Score > −0.5 and NRF2-dependent cell lines are defined as <−0.5. Legend N represents the number of cell lines analyzed per condition. Box represents the first and third quartiles, with the median shown as a horizontal line. Whiskers extend to the smallest and largest value within 1.5 * the interquartile range. Notches extend 1.58 *IQR/sqrt(n). Individual peak Ns - NRF2 Non-dependent: Promoter $n = 2016$, Enhancer $n = 1148$, Linked $n = 2072$. NRF2 dependent: Promoter $n = 792$, Enhancer $n = 451$, Linked $n = 814$. Promoter: $p = 0.029$, Enhancer: $p = 2.63e-05$, Linked: $p = 0.372$. (F) Genome browser snapshots of the group I NRF2 signature gene *GCLC* highlighting the neighboring chromatin landscape. Simulated H3K27ac HiChIP links are colored by p-value. * denotes PRO-seq peak from Fig. 2A. Source data are available online for this figure.

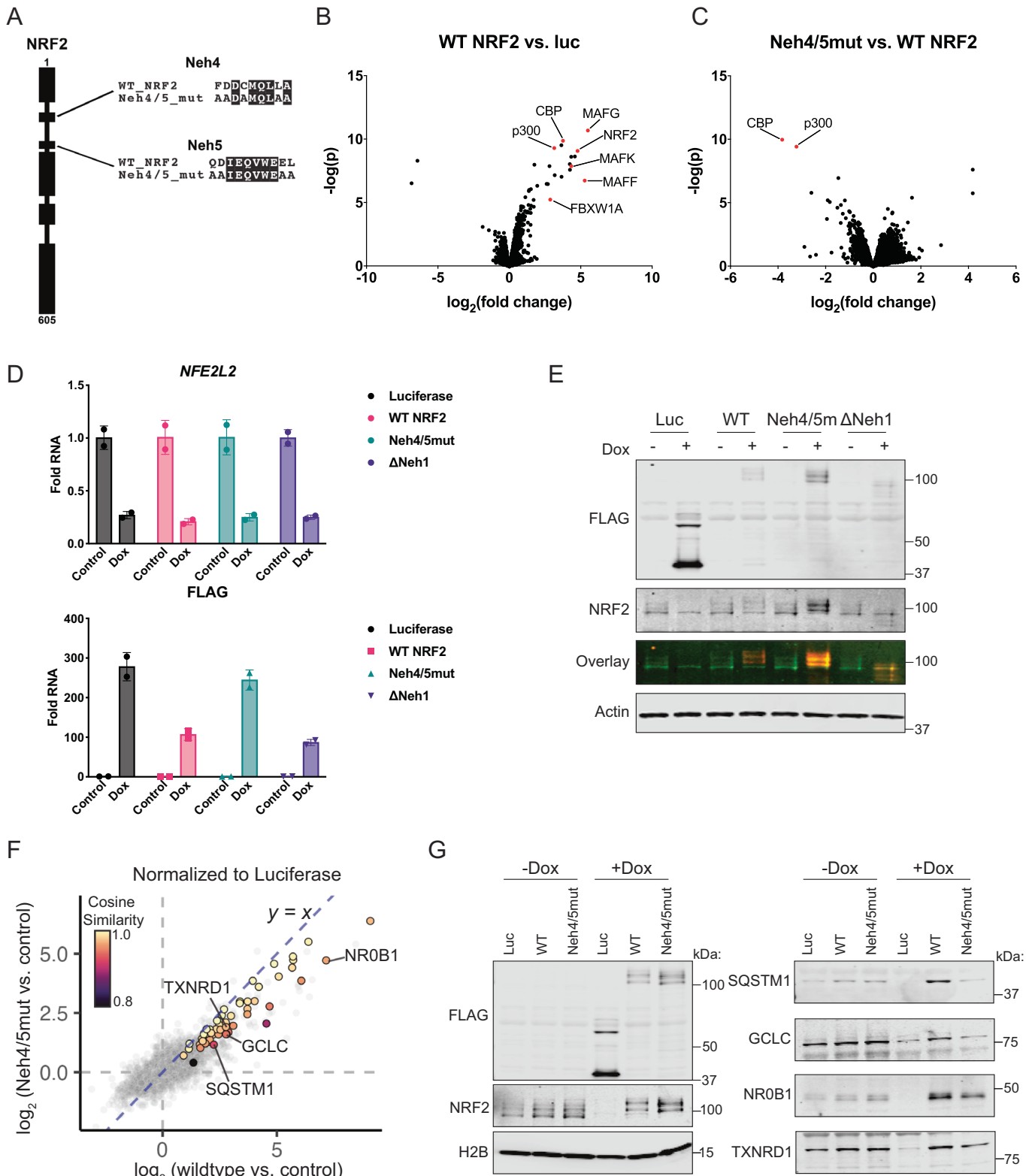

**Figure 3. CBP/p300 is required for robust expression of NRF2 target genes.**

(A) Domain structure of NRF2 indicating point mutations in the Neh4/5mut protein. (B, C) FLAG IP/MS on nuclear fractions from A549 cells expressing Renilla luciferase, WT NRF2, or Neh4/5mut proteins. Differential enrichment between WT NRF2 vs Renilla luciferase (B) and Neh4/5mut vs WT NRF2 (C) are shown. Data are derived from four individual replicates. (D) qRT-PCR from RNA isolated from A549-shNRF2-BIND-luc, -BIND-WT NRF2, -BIND-Neh4/5mut, or -BIND-ΔNeh1 cells treated with water (control) or 2.5 ng/mL dox for 24 h using indicated primers normalized to *RPL13A* as a housekeeping gene. Exogenous NRF2 constructs are codon optimized and thus not detected by *NFE2L2* primer. Data are generated in triplicate with mean +/− SD shown. Representative data from three independent experiments is shown. (E) Western blot of cells from (D) probing with FLAG, NRF2, or β-actin antibodies. Representative data from three independent experiments is shown. (F) Correlation of transcriptomic changes induced by rescue with WT NRF2 vs. Neh4/5mut in A549-shNRF2-BIND-WT NRF2 and -BIND-Neh4/5mut cells after 24 h of dox (2.5 ng/mL) treatment. Values show limma-voom log$_2$ fold changes between 24 and 0 h of treatment for each NRF2 construct, normalized by subtracting the log$_2$ fold change between 24 and 0 h of Luciferase. Highlighted points are NRF2 signature genes colored by cosine similarity between WT and Neh4/5mut. Cosine values of 1 represent the same angle and hence no differential expression between Neh4/5mut and WT NRF2, with more divergence as values decrease. (G) Western blot of cells from A549-shNRF2-BIND-luc, -BIND-WT NRF2 or -BIND-Neh4/5mut cells treated with water (control) or 2.5 ng/mL dox for 48 h, probing for FLAG, NRF2, H2B, SQSTM1, GCLC, NR0B1, and TXNRD1 antibodies. Representative data from three independent experiments is shown. Source data are available online for this figure.

2009), is a validated NRF2 interaction partner, and is known to enhance NRF2 activity at ARE-driven reporter genes (Katoh et al, 2001), we next sought to clarify the functional contribution of CBP/p300 to NRF2-dependent transcription.

NRF2 interaction with CBP is mediated by Neh4 and Neh5 motifs in NRF2 (Katoh et al, 2001). Using homology models (Fig. EV4A) and previous literature (Zhang et al, 2007), we designed a mutant containing four alanine substitutions in both Neh4 and Neh5 (Fig. 3A), and tested this mutant (Neh4/5mut) for interaction with CBP via coimmunoprecipitation. While WT NRF2 robustly pulled down CBP, Neh4/5mut failed to interact with CBP (Fig. EV4B). In contrast, deletion of the NRF2 Neh1 domain (ΔNeh1) retained interaction with CBP (Fig. EV4B). As a control, we blotted for the sMAF protein MAFK, which was bound by both WT and Neh4/5mut, but lost with ΔNeh1, consistent with the known function of Neh1 (Igarashi et al, 1994; Poh et al, 2020) (Fig. EV4B).

To validate the loss of CBP-binding with Neh4/5mut using an unbiased and quantitative approach, we performed immunoprecipitation mass spectrometry (IP/MS) on nuclear extracts isolated from A549 cells expressing FLAG-tagged WT NRF2 or Neh4/5mut, or luciferase as a control. Known interactors were recovered from WT NRF2 pulldowns including all sMAF proteins (MAFF/G/K), FBXW1A (bTrCP-1), and notably both CBP and p300 (Fig. 3B, Dataset EV2). Comparing WT NRF2 to Neh4/5mut, CBP and p300 were the most significantly depleted binding partners in Neh4/5mut pulldowns (Fig. 3C, Dataset EV2), while Neh4/5mut retained wildtype level interactions with sMAFs and FBXW1A (Dataset EV2). These data confirm Neh4/5mut as a specific CBP/p300-deficient NRF2.

To define the direct contributions of CBP/p300 to NRF2 function, we characterized Neh4/5mut in a knockdown / re-expression system. Starting with our A549 shNRF2 cell line (Fig. EV1C–E), we utilized a doxycycline-inducible promoter to acutely express exogenous NRF2 WT, Neh4/5mut, ΔNeh1, or luciferase as a control. Upon dox treatment, we observed similar levels of endogenous NRF2 knockdown and exogenous expression of WT NRF2, Neh4/5mut, ΔNeh1, or luciferase by qPCR and western blotting (Fig. 3D,E). RNA-seq in A549 shNRF2 cells expressing exogenous constructs showed broad attenuation of NRF2 target gene expression when comparing Neh4/5mut to WT (Fig. 3F).

To interpret this difference, we used cosine similarity, a mathematical method to quantify the similarity between two values. Cosine similarity was calculated between the differential

expression of each gene (Neh4/5mut vs. WT NRF2) versus a theoretical 1:1 expression level, with values deviating from 1 signifying divergence (Fig. 3F, Dataset EV3). From these analyses, examples of genes whose expression was most impacted by CBP/p300-deficiency included the group I genes SQSTM1, GCLC, NR0B1, and TXNRD1 (Fig. 3F). Importantly, the observed transcriptional effects translated to differences in protein levels as measured by western blotting (Fig. 3G). We conclude that CBP/p300 is required for robust expression of many NRF2 signature genes.

## CBP/p300 is dispensable for NRF2-dependent cancer cell viability

Apart from NRF2, CBP/p300 activity has been shown to regulate target gene expression of many other oncogenic transcription factors, including certain lineage factors (i.e., AR/ER) as well as oncogenic fusions (i.e., AML1-ETO). Suppressing CBP/p300 activity via A-485, a selective inhibitor of the CBP/p300 acetyltransferase domain, in cancer cells addicted to these transcription factors results in viability defects (Lasko et al, 2017; Waddell et al, 2021; Welti et al, 2021; Zhang et al, 2020b). Provided the similar function of CBP/p300 in regulating NRF2-dependent transcription, we anticipated that CBP/p300 loss-of-function would result in NRF2-specific viability defects.

We first tested NRF2 requirement on CBP/p300 using our knockdown re-expression system comparing WT NRF2, Neh4/5mut, ΔNeh1, or luciferase as a control. All four cell lines had comparable growth kinetics in the absence of dox (Fig. 4A,B). As expected, we observed strong viability defects in both the luciferase expressing line as well as NRF2 ΔNeh1 (consistent with the essential role of Neh1 in binding to both DNA and sMAFs) upon dox addition. Unexpectedly, NRF2 Neh4/5mut showed similar outgrowth compared to NRF2 WT following dox addition (Fig. 4A,B), indicating interaction of CBP/p300 with NRF2 is not required for NRF2 to sustain cell viability. Similar results were obtained in H460 shNRF2 cells (Fig. EV1C–E) engineered with matched constructs (Fig. EV4C–E). Both WT and Neh4/5mut NRF2 rescue cells exhibited comparable outgrowth in a scratch wound healing assay (Fig. EV4F), indicating that CBP/p300 loss-of-function does not impact migratory capacity of cells on tissue culture plates.

To expand upon these observations, we screened A-485 in a 5-day viability assay using a large panel of cancer cell lines from

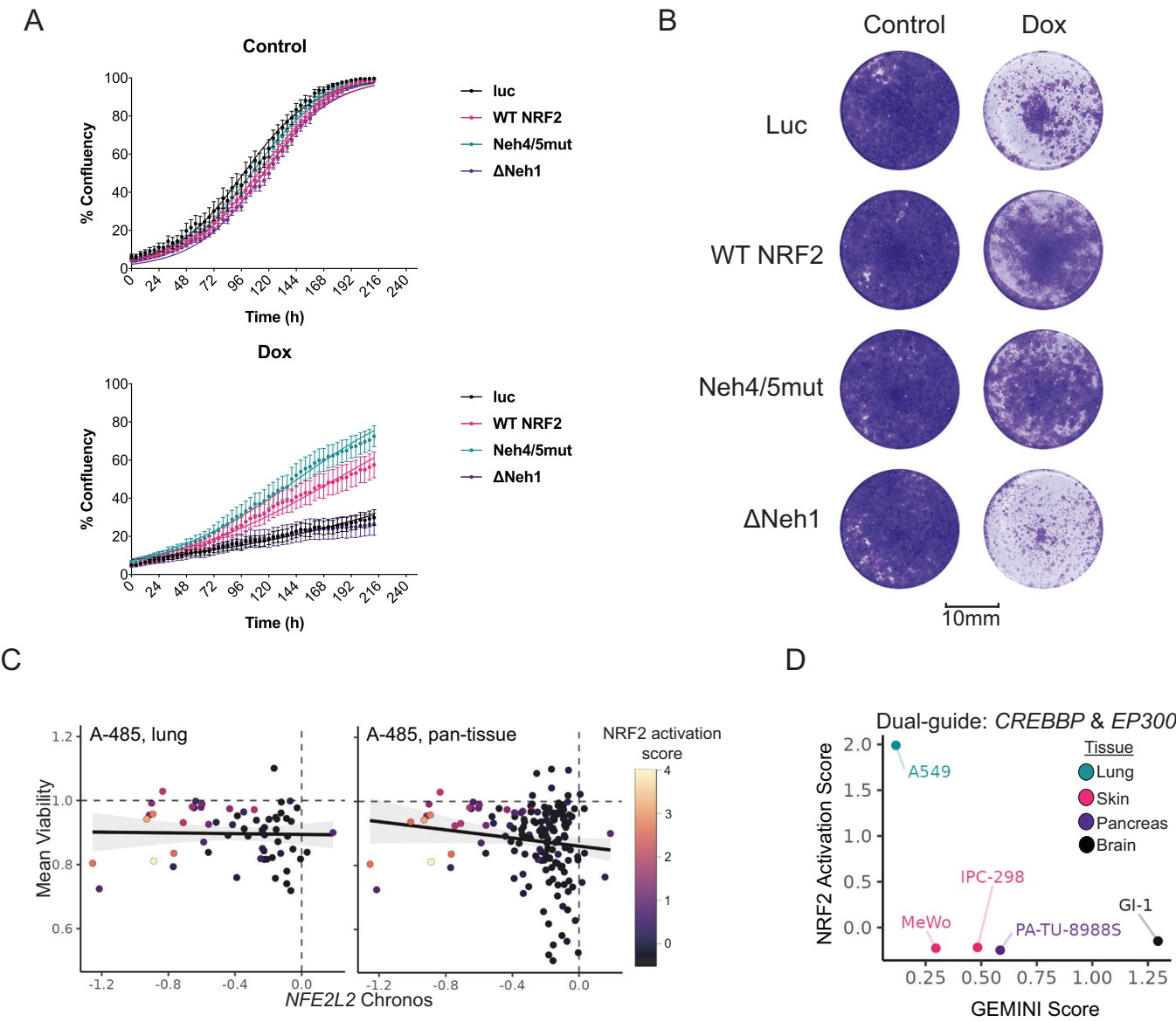

**Figure 4. CBP/p300 is dispensable for NRF2-dependent cancer cell viability.**

(A) Incucyte growth curves of A549-shNRF2-BIND-luc, -BIND-WT NRF2, -BIND-Neh4/5mut, or -BIND-ΔNeh1 cells treated with water (control) or 2.5 ng/mL dox. Representative data from two independent experiments is shown, with mean and SEM derived from 16 images/well indicated. (B) Crystal violet staining of A549-shNRF2-BIND-luc, -BIND-WT NRF2, -BIND-Neh4/5mut, or -BIND-ΔNeh1 cells treated with water (control) or 2.5 ng/mL dox for 8 days. Representative data from three independent experiments is shown. (C) Correlation between A-485 mean viability and NRF2 dependency as defined by *NFE2L2* Chronos score. NRF2 activation score, defined as the average RPKM of NRF2 target genes, is overlaid. Line represents linear regression between mean viability and NRF2 Chronos. (D) NRF2 activation score vs GEMINI Score (Zamanighomi et al, 2019) for co-knockout of *CREBBP* and *EP300*. GEMINI scores represent the added effect of a gene pair interaction, with a score of zero indicating no additional effect beyond each gene's individual contribution. Source data are available online for this figure.

different tissue types. In agreement with previous reports, we observed selective viability defects across the panel of cell lines, with most blood lines and a subset of lines across additional indications showing sensitivity to A-485 treatment (Fig. EV5A) (Lasko et al, 2017). In contrast, control compounds paclitaxel (an anti-mitotic microtubule poison) and pladienolide B (a spliceosome modulator), were indiscriminately toxic in this assay (Fig. EV5A).

We next asked whether there was a correlation between sensitivity to A-485 and NRF2 transcriptional activity (using our transcriptional signature) or NRF2 dependency (Chronos score)

(Dataset EV4). Surprisingly, we observed no correlation (if anything a weak anticorrelation) between mean viability and NRF2 transcriptional activity or NRF2 dependency (Fig. 4C). To further characterize viability effects in an additional assay format, we profiled A-485 (and the inactive analogue A-486) (Lasko et al, 2017) in a panel of NRF2-dependent and NRF2-independent lines using clonogenic assays. Consistent with our previous results, minimal viability effects were observed across the panel, and there was no selective activity in NRF2-dependent cell lines (Fig. EV5B). Lastly, a dual guide approach used to knockout paralogs in tandem

(Ito et al, 2021) demonstrated a clear lack of dependency on *CREBBP* (CBP) and *EP300* (p300) in the NRF2-dependent cell line, A549, with varying degrees of dependency observed in NRF2-independent lines assayed (Fig. 4D). Collectively, these data enforce the finding that CBP/p300 contribution to NRF2 function is not required for the growth of NRF2-dependent cancer cells.

## CBP/p300 requirement for robust GSH biosynthesis and ROS clearance via the NRF2 pathway

Our data establish a direct requirement for NRF2 eRNA synthesis and H3K27ac deposition at enhancers linked to NRF2 target genes, all while CBP/p300 binding and activity are not required for NRF2-dependent cancer cell growth. This led us to question the specific biological significance of CBP/p300-enhanced transcription of NRF2 signature genes.

To interrogate downstream pathways impacted by CBP/p300 deficiency, we performed metabolomics using our knockdown / re-expression system. Comparing dox-treated shNRF2 cells expressing WT NRF2 vs luciferase as a control, we observed 35 metabolites as differentially abundant between the two conditions (Fig. 5A, Dataset EV5). In particular, we observed GSH and several glutamate dipeptides (i.e., Glu-Ala, -Leu, -Cys, -Thr) as significantly depleted upon NRF2 knockdown (Fig. 5A,B,E), in line with the prevailing role of NRF2 in controlling GSH biosynthesis. We also observed disruption of additional metabolites involved in the PPP (sedoheptulose-7-phosphate, S7P), nucleotide (i.e., CMP, IMP) and amino acid (i.e., kynurenine, homocitrulline) metabolism upon NRF2 knockdown (Fig. 5A,B,E).

Unexpectedly, we detected increases in total NADP levels (the other major metabolite commonly associated with NRF2 and identified in our multi-omic analysis, Fig. 1A, Dataset EV5) in response to NRF2 knockdown in our global metabolomics dataset. Provided reported challenges in accurately quantifying this labile metabolite (Hofmann et al, 2010; Lu et al, 2018; Wu et al, 1986), in addition to pathway-level changes incongruous with increased NADP levels (i.e., increased indole-3-lactic acid and decreased nicotinamide riboside, Fig. 5E), we used targeted assays to further interrogate this phenotype. First, we used an optimized LC-MS assay to quantify absolute NADP+ and NADPH levels in A549shNRF2-luc cells upon doxycycline-induced knockdown (Lu et al, 2018). Both NADP+ and NADPH levels were decreased in dox-treated cells (Fig. EV6A,B). To verify that NADPH production is indeed decreased in NRF2 knockdown cells, we performed a stable isotope tracing experiment using [1,2-$^{13}$C]-glucose to determine metabolic flux of PPP, a major NADPH-producing pathway. The PPP flux relative to glycolysis flux in NRF2-knockdown cells was decreased (Fig. EV6C), in line with decreased NADPH levels. To further validate the causality between NRF2 and PPP, we performed the same tracing experiment on H441 cells treated with DMSO and a KEAP1 inhibitor (Davies et al, 2016) inducing NRF2 activation. Relative PPP flux was increased in KEAPi-treated cells (Fig. EV6D). Collectively, our findings confirm that NRF2 positively supports NADP+/H production in our systems.

Focusing on metabolites regulated by CBP/p300, we observed a subset (21 total) that is differentially abundant comparing WT NRF2 to Neh4/5mut (Fig. 5C, Dataset EV5). Of these metabolites, the majority is associated with GSH production and utilization (Fig. 5C–E),

and we confirmed the effect on GSH levels using a targeted quantification assay (Fig. EV6E). Targeted assays measuring GSH/GSSG showed no change in ratios among conditions (Fig. EV6F), demonstrating that differences were attributed to changes in de novo synthesis rather than reduction. Also of note, certain metabolites outside of the GSH pathway were rescued similarly by NRF2 WT and Neh4/5mut (i.e., S7P and CMP, Fig. 5E).

Defects in the GSH biosynthesis should translate to increased ROS and have been recently shown to alter the lipid composition of cells (Asantewaa et al, 2024). To directly test this, we measured $H_2O_2$ in NRF2 knockdown cells rescued with WT NRF2, Neh4/5mut, or luciferase control. NRF2 knockdown markedly increased $H_2O_2$ and was rescued with expression of WT NRF2 (Fig. 5E). In alignment with our GSH data, we observed elevated ROS levels in cells rescued with Neh4/5 compared to rescue by WT NRF2 (Fig. 5F). NRF2 knockdown cells rescued with Neh4/5mut also showed increased sensitivity to the GSH-depleting agent, BSO, compared with cells rescued with WT NRF2 (Fig. EV7A). In further support of the requirement of CBP/p300 for efficient ROS clearance via NRF2, we observed increases in $H_2O_2$ in response to A-485 treatment specifically in NRF2-dependent cell lines and not in NRF2-independent lines (Fig. 5G). Moreover, metabolomics in A549 cells treated with A-485 showed defects in the GSH pathway (Fig. EV7B), consistent with stronger effects on group I genes (including *GCLC*, Fig. 1B) as measured by RNA-seq (Fig. EV7C). In summary, these findings demonstrate that CBP/p300 enhances NRF2-dependent accumulation of glutathione, resulting in suppression of reactive oxygen species in cells.

## CBP/p300-driven glutathione production alters cellular lipid composition, in particular triglycerides

Redox state has been shown to regulate lipogenesis in hepatic function (Brandsch et al, 2010). A recent study has pointed to an emerging role for GCLC/GSH in controlling lipid biosynthesis in mice (Asantewaa et al, 2024), highlighting a significant function for liver-specific expression of *GCLC* in promoting circulating triglyceride (TAG) levels. Given the substantial impairment of GSH production upon blocking CBP/p300 contribution to NRF2-dependent transcription, we next investigated how this may alter the lipid landscape of NRF2-dependent NSCLC cells.

We performed unbiased lipidomics on samples matched to the metabolomics dataset of our NRF2 knockdown / re-expression system. Comparing NRF2 knockdown cells rescued with WT vs luciferase control, we observed broad modulation of lipid composition and abundance, in particular NRF2-dependent increases in TAGs and sterol lipids (cholesteryl esters, CE). We also observed NRF2-dependent decreases in diglycerides (DAGs) and multiple classes of phospholipids (phosphatidylcholine, PC; phosphatidylethanolamine, PE; lysophosphatidylethanolamine, LPE) (Fig. 6A,B, Dataset EV6). Comparing cells rescued with WT NRF2 vs Neh4/5mut, we observed highly restricted lipid abundance changes, with TAGs as the predominant species being driven by CBP/p300 (Figs. 6C,D and EV7, Dataset EV6). These selective changes mirror our metabolomics data, supporting a mechanistic link between GSH and TAGs.

Collectively, these data establish that the functional interaction between CBP/p300 and NRF2 is required for robust antioxidant capacity and TAG biosynthesis, consistent with CBP/p300 regulation of *GCLC* expression.

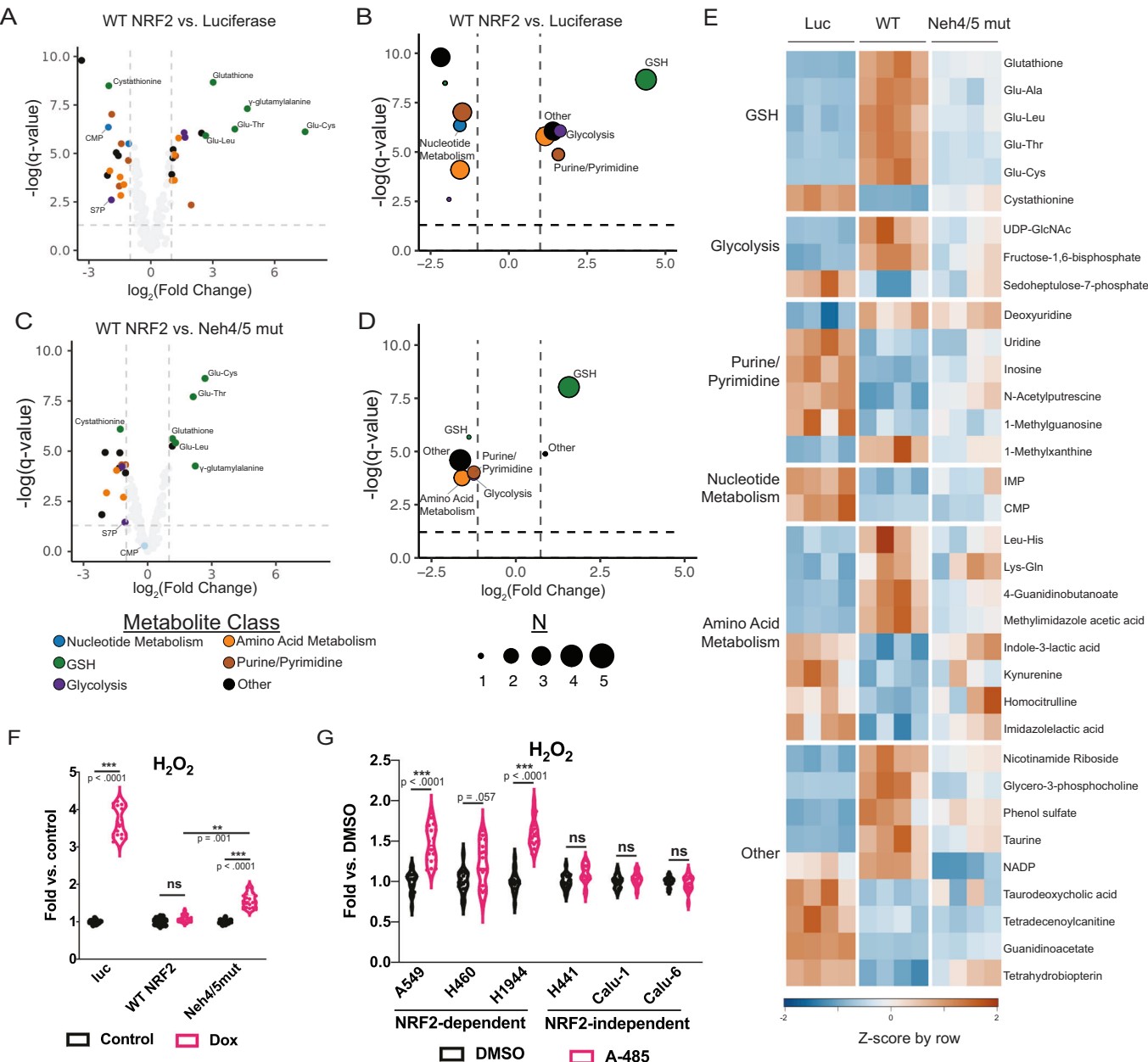

**Figure 5. CBP/p300 is required for robust GSH biosynthesis and ROS clearance via NRF2.**

(A) Volcano plot highlighting the difference in specific metabolites in A549shNRF2 cells rescued with WT NRF2 vs. luciferase control (2.5 ng/mL dox for 48 h). Data are derived from four individual replicates. Log$_2$ fold change values are derived from limma-voom. (B) Volcano plot highlighting the difference in specific metabolite classes in A549shNRF2 cells rescued with WT NRF2 vs. luciferase control (2.5 ng/mL dox for 48 h). Metabolite species are grouped by class and direction of log$_2$ fold change. Q-values for classes are derived from combining individual metabolite statistics from (A) using Tippett's Method. (C) Volcano plot highlighting the difference in specific metabolites in A549shNRF2 cells rescued with WT NRF2 vs. Neh4/5mut (2.5 ng/mL dox for 48 h). Data are derived from four individual replicates. Log$_2$ fold change values derived from limma-voom. (D) Volcano plot highlighting the difference in specific metabolite classes in A549shNRF2 cells rescued with WT NRF2 vs. Neh4/5mut (2.5 ng/mL dox for 48 h). Metabolite species are grouped by class and direction of log$_2$ fold change. Q-values for classes are derived from combining individual metabolite statistics from (C) using Tippett's Method. (E) Heatmap of significantly different metabolites with color intensity representing fold dox vs. control normalized by row as a z score. Each column represents an individual replicate. (F) H$_2$O$_2$ levels in A549shNRF2 cells rescued with luciferase control, WT or Neh4/5mut NRF2 (2.5 ng/mL dox for 48 h). Data represent three independent experiments performed as sextuplicates. P-values calculated by an ANOVA followed by Tukey's HSD post-hoc test, luc – (dox vs. control) $p = 6.611472e-10$, Neh4/5mut (dox vs. control) $p = 6.611472e-10$, WT vs Neh4/5mut (dox vs dox) $p = 1.196241e-03$ (G) H$_2$O$_2$ levels across cells treated with 1 µM A-485 for 24 h. Data represent three independent experiments performed as sextuplicates. P-values determined by independent two-sample t-tests between DMSO and A-485 treatment for each cell line, followed by Bonferroni correction for multiple testing. A549 $p = 3.966883e-08$, H460 $p = 5.778530e-02$, H1944 $p = 1.322294e-09$. Source data are available online for this figure.

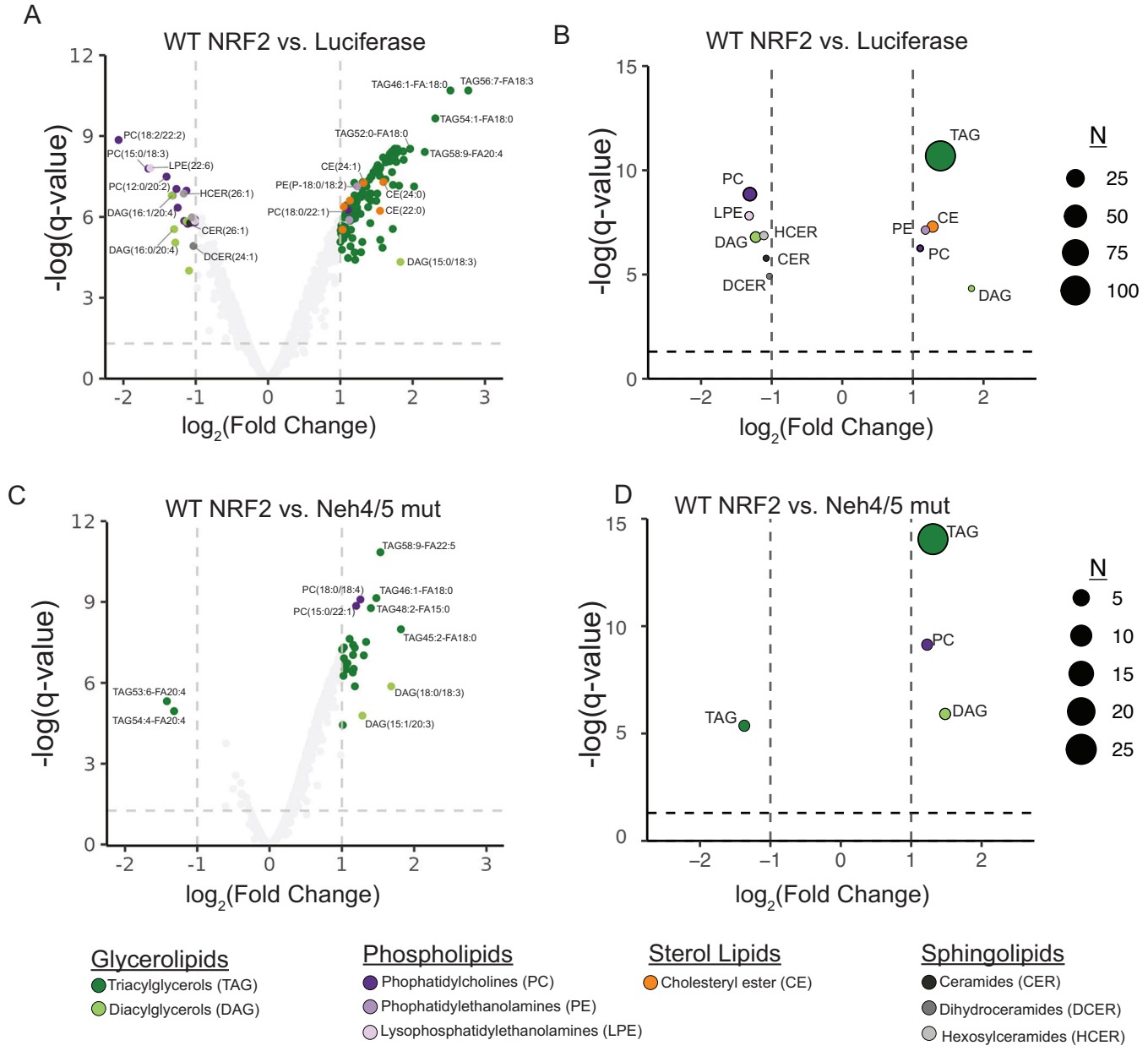

**Figure 6. CBP/p300-driven glutathione production alters cellular lipid composition, in particular triglycerides.**

(A) Volcano plot highlighting the difference in specific lipids in A549shNRF2 cells rescued with WT NRF2 vs. luciferase control (2.5 ng/mL dox for 48 h). Data are derived from four individual replicates. (B) Volcano plot highlighting the difference in lipid classes in A549shNRF2 cells rescued with WT NRF2 vs. luciferase control (2.5 ng/mL dox for 48 h). Lipid species are grouped by class and direction of Log2 fold change. Q-value for classes derived from Tippett's Method. Data are derived from four individual replicates. (C) Volcano plot highlighting the difference in specific lipids in A549shNRF2 cells rescued with WT NRF2 vs. Neh4/5mut (2.5 ng/mL dox for 48 h). Data are derived from four individual replicates. Log2 fold change values derived from limma-voom. (D) Volcano plot highlighting the difference in lipid classes in A549shNRF2 cells rescued with WT NRF2 vs. Neh4/5mut (2.5 ng/mL dox for 48 h). Data are derived from four individual replicates. Q-values for classes are derived from combining individual metabolite statistics from (C) using Tippett's Method. Source data are available online for this figure.

## Discussion

While the prevailing role of NRF2 in normal physiology is to transiently transactivate antioxidant genes in response to specific stressors, deciphering the pleiotropic functions of constitutive NRF2 activity in cancer has been the focus of extensive efforts (Rojo de la Vega et al, 2018). Through comprehensive genomic characterization of genes driving NRF2-dependent cancer cell proliferation, we interrogated the functional interaction between NRF2 and CBP/p300 to uncover a role for this interaction in driving antioxidant capacity. We report that suppressing GSH synthesis and inducing ROS through CBP/p300 loss-of-function paradoxically does not impact NRF2-dependent cancer cell proliferation. It is becoming increasingly clear that additional

metabolic functions for NRF2 exist in regulating amino acid, lipid, and nucleic acid biosynthetic pathways, and our work highlights the importance of these broader roles of NRF2 upon chronic activation in cancer.

Although disrupting the function of CBP/p300 broadly attenuated expression of NRF2 target genes (Fig. 3F), select, highly-enhancer driven genes were more reliant on CBP/p300. In agreement with our metabolomic and lipidomic datasets showing depletion of GSH metabolites and TAGs following rescue by CBP/p300-deficient NRF2, CBP/p300 was required for robust expression of *GCLC*. The finding that proliferation of NRF2-dependent cancer cells was unaffected by these considerable transcriptional, metabolomic, and lipidomic disruptions in response to CBP/p300 loss-of-function implies that the classical function of NRF2 in enhancing antioxidant capacity does not fully explain its role in cancer cell maintenance. This is in agreement with published findings that GSH and other ROS scavengers did not rescue NRF2 knockdown, despite diminishing ROS levels (Vartanian et al, 2019). Similarly, Vartanian et al showed that increased ROS resulting from SLC7A11 inhibition did not impact viability of *KEAP1* mutant cancer cells. These and our results point to alternative NRF2 functions as being essential for cancer cell growth (i.e., PPP, nucleotide biosynthesis (Mitsuishi et al, 2012), amino acid metabolism (DeNicola et al, 2011), etc.). Interestingly, a recent report identified disruptions in nucleotide biosynthesis as a potential cause of the selective toxicity of class I HDAC inhibition toward NRF2-dependent cancer cells (Karagiannis et al, 2024). We note mainly elevated levels of nucleotide metabolites in NRF2 knockdown cells that are efficiently rescued by WT and Neh4/5mut NRF2 (Fig. 5E). Future studies will be focused on the biological relevance of these nucleotide disruptions in connection with the PPP and amino acid biosynthetic pathways. Also of importance, and beyond the scope of the current 2D cell culture assays (Figs. 4 and EV4), is understanding the impact of CBP/p300-enhanced NRF2 redox signaling on cancer cell invasion/migration within the framework of metastatic disease progression, ultimately in in vivo systems.

CBP/p300 has long been assumed to be an essential coactivator of NRF2-dependent transcription via direct interaction with NRF2 (Katoh et al, 2001). However, much of this functional characterization utilized non-physiological systems with engineered ARE-driven reporters. These studies, along with ChIP-seq analysis highlighting a high density of ARE elements within NRF2 target gene promoters (Namani et al, 2019), led to a promoter-centric model of CBP/p300 coactivation of NRF2-dependent gene expression (Tonelli et al, 2018). Here, we addressed the location and significance of genome-wide NRF2 binding. We define a set of NRF2 target genes that are devoid of promoter-proximal NRF2, calling into question canonical mechanisms of NRF2-dependent transcription. Our multi-omic analysis established a broad role of NRF2 in eRNA transcription and H3K27ac deposition by binding to proximal and distal enhancer elements linked to NRF2 signature genes. Our data indicate that NRF2 acts upstream of eRNA synthesis and CBP/p300, in agreement with the predictive rather than functional role of H3K27ac in regulating transcription (Sankar et al, 2022; Zhang et al, 2020a).

The finding that NRF2-dependent cancer cells are refractory to CBP/p300 inhibition distinguishes NRF2 from other transcriptional oncogene addictions (i.e., AR, BRD4-NUT). Reasons underlying this differential coactivator dependency are currently unclear, but

we speculate that the direct role for NRF2 in enhancer function described here may explain why NRF2-dependent cancer cells are insensitive to CBP/p300 inhibition. For example, NRF2 may be required for the generation and maintenance of enhancer hubs (Zhao and Faryabi, 2023) that function upstream of CBP/p300 activity. A previous report linked NRF2 function at enhancers to a complex containing C/EBPβ (Okazaki et al, 2020) and while we detected this interaction in our nuclear interactomics experiment, it did not meet significance criteria (Dataset EV2). Our dataset did, however, uncover several novel NRF2 interaction partners implicated in transcriptional regulation (i.e., SMARCE1, PTOV1, CHCHD2). Dissecting the composition and function of NRF2 complexes present at canonical ARE-driven promoters vs. proximal/linked enhancers will be the subject of future investigations.

In summary, this work establishes that NRF2 has direct functions at enhancers and is sufficient to drive cancer cell growth in the absence of CBP/p300-enhanced glutathione synthesis. We anticipate that our work will inform future studies on the complex functions of NRF2 in cancer, and ultimately aid in efforts to therapeutically target this pathway for the benefit of cancer patients.

# Methods

**Reagents and tools table**

| Reagent/Resource | Reference or Source | Identifier or Catalog Number |
|---|---|---|
| **Experimental models** | | |
| A549 | Genentech Cell Bank (Yu et al, 2015) | N/A |
| NCI-H460 | Genentech Cell Bank (Yu et al, 2015) | N/A |
| HEK293 | Genentech Cell Bank (Yu et al, 2015) | N/A |
| **Recombinant DNA** | | |
| EF1α-/TRE3G-promoter-driven Renilla luciferase and NRF2 (WT NRF2, Neh4/5mut, SV40-NLS-ΔNeh1) constructs | GenScript/this work | N/A |
| GFP-tagged CBP | Karen Gascoigne (Raisner et al, 2018) | N/A |
| **Antibodies** | | |
| FLAG | Sigma | F3165 |
| NRF2 | Abcam | ab62352 |
| H2B | CST | 12364 |
| SQSTM1 | CST | 88588 |
| GCLC | CST | 48005 |
| TXNRD1 | Abcam | ab16840 |
| NR0B1 | CST | 13538 |

| Reagent/Resource | Reference or Source | Identifier or Catalog Number |
|---|---|---|
| MAFK | Abcam | ab229766 |
| CBP | CST | 7389 |
| β-actin | CST | 3700 |
| **Oligonucleotides and other sequence-based reagents** | | |
| NFE2L2 | Thermo Fisher | Hs00975961_g1 |
| RPL13A | Thermo Fisher | Hs04194366_g1 |
| FLAG | IDT/this work | N/A |
| **Chemicals, Enzymes and other reagents** | | |
| A-485 | MedChemExpress | HY-107455 |
| Paclitaxel | Sigma-Aldrich | 580556 |
| Pladienolide B | Tocris | 6070 |
| Doxycycline | Takara Bio | 631311 |
| BSO | MedChemExpress | HY-106376A |
| Puromycin | Fisher/Gibco | A1113803 |
| KEAP1 inhibitor | Wuxi | Davies et al, 2016 |
| RIPA buffer | G Biosciences | 786-490 |
| Halt Protease Inhibitor | Thermo | 78446 |
| NuPAGE™ LDS Sample Buffer (4X) | Thermo | NP0007 |
| NuPAGE™ Sample Reducing Agent (10X) | Thermo | NP0009 |
| WedgeWell™ Tris-Glycine protein gels | Thermo | XP04205BOX |
| NuPAGE™ Tris-Acetate Mini Protein Gels | Thermo | EA03755BOX |
| Precision Plus Protein™ Kaleidoscope™ Prestained Protein Standards | Bio-Rad | 1610375 |
| Trans-Blot Turbo Midi 0.2 µm Nitrocellulose Transfer Packs | Bio-Rad | 1704159 |
| Intercept® (PBS) Blocking Buffer | Licor | 927-70010 |
| High-Capacity cDNA Reverse Transcription Kit | Thermo | 4374966 |
| Lipofetamine 3000 | Thermo | L3000015 |
| Crystal Violet Solution | Sigma | HT90132 |
| BCA Protein Assay Kit | Pierce | A65453 |
| CellTiter-Glo Luminescent Cell Viability Assay | Promega | G7571 |
| ROS-Glo H₂O₂ Assay | Promega | G8821 |
| RNeasy Plus Mini Kit | Qiagen | 74136 |

| Reagent/Resource | Reference or Source | Identifier or Catalog Number |
|---|---|---|
| TaqMan™ Gene Expression Master Mix | Thermo | 4369016 |
| Qubit RNA HS Assay Kit | Thermo | Q32852 |
| Truseq Stranded mRNA kit | Illumina | 20020595 |
| Qubit dsDNA HS Assay Kit | Thermo | Q32851 |
| Anti-FLAG® M2 Magnetic Beads | Sigma | M8823 |
| Lysyl endopeptidase Lys-C | Fujifilm Wako | 4987481427648 |
| Trypsin | Promega | VA9000 |
| Sep-Pak C18 cartridges | Waters Corporation | WAT051910 |
| High pH Reversed-Phase Peptide Fractionation Kit | Pierce | 84868 |
| Aurora Ultimate 25 cm × 75 µm BEH C18 1.7 µm | IonOpticks | AUR3-25075C18 |
| ACQUITY UPLC BEH Amide Column | Waters Corporation | 186004802 |
| ¹⁵N₅-ADP | Sigma | 741167 |
| [1,2-¹³C]-glucose | Sigma | 453188 |
| **Software** | | |
| Genedata Screener®: Version 15 | Genedata | N/A |
| Polly platform | Elucidata Corporation | N/A |
| MSStats | Choi et al, 2014 | https://doi.org/10.18129/B9.bioc.MSstats |
| HTSeqGenie | *HTSeqGenie: A NGS analysis pipeline*. R package version 4.35.0. | https://doi.org/10.18129/B9.bioc.HTSeqGenie |
| Spotfire® Dashboard | TIBCO | N/A |
| Chromeleon | Thermo | N/A |
| Lipidyzer™ Platform | Sciex | N/A |
| Bowtie2 | | https://bowtie-bio.sourceforge.net/bowtie2/index.shtml |
| samtools | | https://www.htslib.org/ |
| Picard | | https://broadinstitute.github.io/picard/ |
| MACS2 | | https://www.biorxiv.org/content/10.1101/496521v1 |
| DiffBind | | http://bioconductor.org/packages/release/bioc/vignettes/DiffBind/inst/doc/DiffBind.pdf) |
| FitHiChIP | Bhattacharyya et al, 2019 | https://doi.org/10.1038/s41467-019-11950-y |

| Reagent/Resource | Reference or Source | Identifier or Catalog Number |
|---|---|---|
| GenomicRanges R package | Lawrence et al, 2013 | https://bioconductor.org/packages/release/bioc/html/GenomicRanges.html |
| bedtools | | https://bedtools.readthedocs.io/en/latest/ |
| pyGenomeTracks | Lopez-Delisle et al, 2021 | https://pygenometracks.readthedocs.io/en/latest/ |
| Wiggletools | Zerbino et al, 2014 | https://github.com/Ensembl/WiggleTools |
| Bioconductor | Huber et al, 2015 | https://www.bioconductor.org/ |
| GSNAP | Wu and Nacu, 2010; Wu et al, 2016 | |
| limma-voom | Law et al, 2014 | |
| Latent Semantic Analysis R package | | https://doi.org/10.32614/CRAN.package.lsa |
| Wound Healing Size Tool | Suarez-Arnedo et al, 2020 | |
| ImageJ/Fiji | Schindelin et al, 2012 | |

## Cell culture & western blotting

All cell lines were obtained from the Genentech cell bank (Yu et al, 2015) and grown in RPMI or DMEM supplemented with 10% FBS, 1% glutamine, and 1% penicillin-streptomycin at 37 °C, 5% $CO_2$. Cell lines were confirmed via STR profiling and were mycoplasma free.

For western blotting of cell lysates, exponentially growing cells were lysed with RIPA buffer (G Biosciences) + 1X Halt (Thermo) and centrifuged for 5 min at 10,000 rpm at 4 °C. Lysates were sonicated using the Active Motif EpiShear™ Probe Sonicator under the following conditions: 30% amplitude, 10 s on, 2 s off, for 20 s per sample. Cleared lysates were centrifuged for 5 min at 10,000 rpm at 4 °C and the resultant supernatants were decanted and quantified by BCA assay (Pierce). Protein concentrations of lysates were normalized in RIPA, diluted to 2X in NuPAGE™ LDS Sample Buffer (4X, Thermo) + NuPAGE™ Sample Reducing Agent (10X, Thermo), boiled at 70 °C for 5 min and centrifuged for 5 min at 10,000 rpm. Denatured samples were run on WedgeWell™ Tris-Glycine protein gels (Thermo) with Precision Plus Protein™ Kaleidoscope™ Prestained Protein Standards (Bio-Rad) and transferred to nitrocellulose membranes using the Transblot Turbo Transfer system (Bio-Rad). For CBP blots, NuPAGE™ Tris-Acetate Mini Protein Gels (Thermo) were used. Blocking, primary and secondary antibody binding were performed as standard technique using the Intercept® (PBS) Blocking Buffer (Licor) and membranes were imaged on the Licor CLx Imager.

## Chemicals, antibodies, primers

Commercial chemicals: A-485 (MedChemExpress, Cat#HY-107455), Paclitaxel (Sigma-Aldrich, Cat#580556), Pladienolide B (Tocris, Cat#6070), Doxycycline (Takara Bio, Cat#631311), BSO (MedChemExpress, Cat#HY-106376A) and puromycin (Fisher/Gibco, Cat#A1113803).

Primers (Thermo or IDT): NFE2L2 (Hs00975961_g1), FLAG (custom, below), ABCC2 (Hs00960489_m1), ME1 (Hs00159110_m1), OSGIN1 (Hs00203539_m1), SLC7A11 (Hs00921938_m1), TRIM16L (Hs02598492_mH), TXNRD1 (Hs00917067_m1), RPL13A (Hs04194366_g1).

A custom FLAG primer synthesized at IDT:
Forward: AGG ATG ACG ACG ATA AGG ACT ATA A
Rev: CCC GCG CTG CCT TTA TC
PRB: /5HEX/TCA TCT TTG /ZEN/TAG TCC TTG TCA TCA TCG TCC /3IABkFQ/

Antibodies: FLAG (Sigma Cat#F3165), NRF2 (Abcam Cat#ab62352), H2B (CST Cat#12364), SQSTM1 (CST Cat#88588), GCLC (CST Cat#48005) TXNRD1 (Abcam Cat#ab16840), NR0B1 (Cat#13538), MAFK (Abcam Cat#ab229766), CBP (CST Cat#7389), β-actin (CST Cat#3700).

## Plasmids

Puromycin-resistance containing, either EF1α- or TRE3G-promoter-driven Renilla luciferase and NRF2 (WT, Neh4/5mut, SV40-NLS-ΔNeh1, codon optimized)-T2A-BFP constructs were synthesized at GenScript. GFP-tagged CBP was a generous gift of Karen Gascoigne, cloned in a manner analogous to p300 constructs described in (Raisner et al, 2018).

## Cell line generation & growth assays

A549 and H460 cells were transduced with a GFP-selectable shNTC or shNRF2 virus derived from Vartanian et al, 2019 and single cell cloned using standard protocols. A549-BIND, A549shNRF2-BIND, and H460shNRF2-BIND cell lines were generated using the Piggybac system. Donor constructs were co-transfected with transposase (Ding et al, 2005) at a 4:1 ratio using Lipofectamine 3000 (Thermo) according to manufacturer's protocol. Three days post-transfection, puromycin selection (1 μg/mL) was initiated and was performed for ~2 weeks prior to experimentation. BIND transfected cells represent polyclonal pools.

Cell viability was measured using CellTiter-Glo® Luminescent Cell Viability Assay (Promega) using the Envision 2103 Plate Reader (Promega). Cell growth curves were generated using the IncuCyte System (Sartorius) according to the manufacturer's protocol. Curve fits were performed in Prism. Clonogenic assays were performed by washing cells in PBS, staining with 0.5% crystal violet solution (Sigma HT90132) for 5 min, and washing 3x with PBS.

## RNA isolation, reverse transcription, qRT-PCR, and RNA-seq

RNA isolation was performed using the RNeasy Plus Mini Kit (Qiagen) and reverse transcription was performed using the High-Capacity cDNA Reverse Transcription Kit (Thermo) according to the manufacturer's protocol. qRT-PCR was performed using Taqman primers (Thermo, cat# listed above) and the QuantStudio™ 7 Flex Real-Time PCR System (Thermo). qRT-PCR data were analyzed using the ddCt method.

For RNA-seq studies, total RNA was quantified with Qubit RNA HS Assay Kit (Thermo Fisher Scientific) and quality was assessed

using RNA ScreenTape on TapeStation 4200 (Agilent Technologies). For sequencing library generation, the Truseq Stranded mRNA kit (Illumina) was used with an input of 100–1000 ng of total RNA. Libraries were quantified with Qubit dsDNA HS Assay Kit (Thermo Fisher Scientific) and the average library size was determined using D1000 ScreenTape on TapeStation 4200 (Agilent Technologies). Libraries were pooled and sequenced on NovaSeq 6000 (Illumina) to generate 30 million single-end 50-base pair reads for each sample.

## Immunoprecipitations

Exponentially growing 293 cells ($\sim 8 \times 10^6$) were transfected with 5 µg total DNA of indicated constructs using Lipofectamine 3000 (Thermo) according to the manufacturer's protocol. Three days post-transfection, cells were lysed in FLAG-IP buffer (50 mM Tris-HCl, 150 mM NaCl, 1 mM EDTA, 1% Triton X-100, and 1X Halt). Lysates were sonicated and decanted as described above. 2% was retained as input, and ~2 mg of lysate was subjected to FLAG immunoprecipitation using Anti-FLAG® M2 Magnetic Beads (Sigma-Aldrich, 50 µl/IP washed in FLAG-IP Buffer) overnight at 4 °C. IPs were washed 3X with FLAG-IP buffer, and eluted using 2X in NuPAGE™ LDS Sample Buffer (4X, Thermo) + NuPAGE™ Sample Reducing Agent (10X, Thermo). Western blotting was performed as described above.

For IP/MS experiments on nuclear lysates, A549 cells were stably transfected with TRE3G-driven Renilla luciferase, WT NRF2 or Neh4/5mut as described above. 24 h post-induction with 1 ng/mL (luc), 10 ng/mL (FL), or 5 ng/mL (Neh4/5mut) doxycycline, cells were trypsinized, washed with ice-cold PBS, and resuspended in DR Buffer A (10 mM HEPES-KOH, 10 mM KCl, 1.5 mM MgCl$_2$, 0.5 mM DTT, supplemented with 1X Halt) at a density of $2 \times 10^7$ cells/mL. Resuspensions were incubated on ice for 10 min, followed by outer membrane lysis by 10 strokes of a Dounce tight pestle (Wheaton). Nuclei were sedimented at 10,000 rpm at 4 °C for 10 min and the supernatant was discarded. Nuclei were resuspended in 1 mL FLAG-IP Buffer, sonicated as described above, and cleared by centrifugation, with the resulting supernatant treated as the nuclear lysate. Lysate inputs were normalized among conditions to 8 mg per IP, and immunoprecipitations were performed using Anti-FLAG® M2 Magnetic Beads (Sigma-Aldrich, 50 µl/IP washed in FLAG-IP Buffer) overnight at 4 °C. IPs were washed 3X with FLAG-IP buffer, and 1X with PBS prior to processing for MS. Data are derived from four individual replicates of dox-treated cells.

## On-bead digestion and tandem mass tag (TMT) sample preparation

On-bead trypsin digestion was performed manually with the following protocol. Beads were washed twice with 25 mM ammonium bicarbonate. Protein samples were reduced with dithiothreitol (10 mM, 1 h, 60 °C) and alkylated in the dark with iodoacetamide (15 mM, 30 min, 25 °C). Supernatants were removed and beads/proteins digested with 20 ng lysyl endopeptidase Lys-C (Fujifilm Wako) at 37 °C for 4 h, followed by addition of 100 ng trypsin (Promega) with incubation at 37 °C for 18 h. Digestion was quenched with formic acid and supernatant were desalted via solid-phase extraction with SepPak c18 cartridges (Waters Corp). Peptides were taken to dryness using a lyophilizer. TMTpro 16plex™ (ThermoFisher

Scientific) labeling of peptides were as follows: 20 µL of acetonitrile was added to each TMT tag tube and mixed aggressively. Tags were incubated at room temperature for 15 min. Lyophilized peptides were reconstituted with 100 µL of 50 mM HEPES pH 8.5. 20 µL of label was added to each peptide sample and mixed aggressively. Samples were incubated in an Eppendorf Thermomixer at 300 rpm 25 °C for 1 h. Reactions were terminated with the addition of 5 µL of fresh 5% hydroxylamine solution and 15 min incubation at room temperature. Each labeled sample was pooled, frozen, lyophilized and subjected to SPE on a Waters SepPak 3cc 200 mg c18 cartridge. The eluent was lyophilized.

## TMT peptide fractionation and mass spectrometry

Peptides were fractionated using the High pH Reversed-Phase Peptide Fractionation Kit (Pierce) according to manufacturer's instructions to 16 fractions then combined to 8 fractions in the following order (fractions $1 + 9$, $2 + 10$, $3 + 11$, $4 + 12$, $5 + 13$, $6 + 14$, $7 + 15$, $8 + 16$) for equal distribution. Fractions lyophilized and reconstituted in 0.1% TFA. Peptides (50% per fraction) were analyzed by nano LC/MS/MS Thermo Scientific Dionex UltiMate 3000 UHPLC system interfaced to a ThermoFisher Orbitrap Eclipse™ Tribid™ mass spectrometer. Peptides were analyzed over an Aurora Ultimate 25 cm × 75 µm BEH c18 1.7 µm resin analytical column at 300 nL/min (IonOpticks, Australia). A 180 min gradient was employed per fraction. The mass spectrometer was operated using a custom SPS-MS3 method with FAIMS (@ −50 V and −70 V) and Thermo "Real-Time-Search" against Human Uniprot.org proteome FASTA database with protein-close-out 5 enabled. MS scans were acquired in the Orbitrap at 120,000 FWHM resolution, MS2 scans were acquired in the ion trap using CID at 30% NCE, product ions were isolated using synchronized precursor selection (SPS) and fragmented using HCD at 40% NCE. The isolation window and number ions for SPS-MS3 was adjusted based on the charge state of the precursor. MS3 scans were acquired in the Orbitrap at 50,000 FWHM resolution from $m/z$ 100. A 1 s cycle time was employed for all steps. Data were processed via in-house software and searched COMET (Eng et al, 2013) algorithm against UniProt.org Human taxonomy database with appropriate modification parameters. Searched data were filtered with 1% peptide false-discovery-rate (FDR). TMT quantitative results were processed with in-house software Mojave then results were compiled and visualized with custom Spotfire® Dashboard (TIBCO). TMT data analysis and statistics were done using the MSStats (Choi et al, 2014) package for R.

## Cellular compound screening

Cells were maintained in RPMI-1640, 5% fetal bovine serum, and 2 mM glutamine in a humidified incubator maintained at 37 °C with 5% CO$_2$. Cells were assessed with a Vi-CELL Cell Viability Analyzer (Beckman Coulter); viability of at least 90% was required for screening. A Multidrop™ Combi Reagent Dispenser (Thermo Scientific; Waltham, MA) was used to plate 1000 cells into Falcon® 384-well, black, clear-bottom plates (Catalog No. 353962; Corning). A 9-point dose titration of compounds at 1000x was prepared in advance using an Echo 555 liquid handler (Labcyte) and stored at −80 °C until the day of compound addition. On the following day, cells were treated with a 9-point dose titration of the specific

compound using a Bravo Automated Liquid-Handling Platform (Agilent). After 5 days, 25 μL CellTiter-Glo® reagent was added using a MultiFlo™ Microplate Dispenser (BioTek). Cell lysis was induced by mixing for 30 min on an orbital shaker; plates were then incubated at room temperature for 10 min to stabilize the luminescent signal. Luminescence was read by a 2104 EnVision® Multilabel Plate Reader (PerkinElmer).

Data was processed using Genedata Screener®, Version 15 (Genedata; Basel, Switzerland), with a four-parameter Hill equation using compound dose−response data normalized to the median of 42 vehicle-treated wells on each plate. A "Robust Fit" strategy was also employed by Genedata Screener®, which is based on Tukey's biweight and is resistant to outlier data. The mean fitted viability across the nine tested doses (i.e., area under the viability curve) was computed.

## Metabolomics analysis

Four replicates of 20–30 million A549shNRF2-BIND-luc, WT NRF2, or Neh4/5mut cells were treated with 2.5 ng/mL dox for 48 h, upon which cells were trypsinized, washed 3x with ice-cold PBS and flash frozen in liquid nitrogen for metabolic quenching. All metabolite extraction procedures were kept on ice. Briefly, 350 μL of cold methanol containing in-house metabolomics Recovery IS (Stable Isotope Labeled Internal Standards) mixture and 200 μL of cold chloroform were added to each sample. Samples were vortexed for 10 s, homogenized with beads for 2 min, and centrifuged at 4000 RPM for 5 min. Next, supernatants were transferred and 200 μl cold water was added to perform liquid–liquid phase separation. Samples were mixed and centrifuged again. The top layer aliquots were transferred, dried and reconstituted in 100 μl acetonitrile:water (8:2, v/v) containing the in-house metabolomics Global IS (Stable Isotope Labeled Internal Standards) mixture and submitted for metabolomics analysis. For global metabolomics, ACQUITY UPLC BEH Amide Column (2.1 mm × 150 mm × 1.7 μm, 130 Å, Waters Corporation) was used to separate metabolites with mobile phase A of 100% water containing 10 mM ammonium formate and 0.125% formic acid, and mobile phase B of 95% acetonitrile in water containing 10 mM ammonium formate and 0.125% formic acid. Data acquisition was achieved on a Shimadzu Nexera HPLC series system (Shimadzu, Kyoto, Japan) coupled with a Thermo Q Exactive Plus Hybrid Quadrupole-Orbitrap Mass Spectrometer (Thermo Fisher Scientific). Injection volume of 3 μL was used for sample analysis under Heated Electrospray Ionization (HESI) condition. Samples were run for both positive and negative modes. The Q Exactive Plus Mass Spectrometer was operated with the following parameters: Sheath gas flow rate, 50 units; Aux gas flow rate, 13 units; Aux gas temperature, 425 °C; Capillary temperature, 263 °C; Spray voltage, 3500 V for pos and −2500 V for neg; Scan mode, Full MS scan with data-dependent MS/MS acquisition. In Full MS scan, scan range is 60–900 $m/z$; resolution is 70,000; AGC target, $1 \times e^6$; Maximum IT, 200 ms. In ddMS2 scan, top 5 ions are selected with an isolation window of 1.5 $m/z$; resolution is 17,500; AGC target, $5 \times e^4$; Maximum IT, 20 ms. Data processing software Chromeleon (Thermo Fisher Scientific), and Polly platform (Elucidata Corporation) were used for metabolite identification, peak picking/peak integration, and statistical analysis, respectively. Metabolites were identified at Level 1 confidence by matching at least two

independent orthogonal experimental data (accurate mass, isotopic ratio, retention time, and MS/MS fragmentation pattern) against in-house compound RT library. Trend analysis of stable isotope labeled internal standards and matrix PoolQC samples were examined (with %RSD less than 15%) to validate system suitability and data robustness. For each metabolite, relative quantification was obtained through either MS peak area of analyte or MS peak area ratio of analyte/internal standard.

## H₂O₂ and GSH measurements

A549-shNRF2-BIND-luc, BIND-WT, or BIND-Neh4/5mut NRF2 cells were treated +/−2.5 ng/mL dox for 48 h. Total GSH levels were measured using the GSH/GSSG-Glo™ Assay (Promega) according to the manufacturer's protocol. $H_2O_2$ was measured using the ROS-Glo™ $H_2O_2$ Assay (Promega) according to the manufacturer's protocol. For both assays, raw relative light units (RLU) values were normalized against CellTiter-Glo (CTG) values, with RLU/CTG values for each cell line expressed as a fold relative to untreated cells.

## Scratch wound healing assay

A549shNRF2-BIND-luc, WT, Neh4/5mut, ΔNeh1 cells were treated +/−2.5 ng/mL dox for 24 h, followed by scratching using a cell scraper. Cells were washed 3X with PBS, media (+/−2.5 ng/mL) was replaced, and cells were subjected to Incucyte imaging for 7 days with media/dox refreshed every 2–3 days. Scratch healing rate was quantified using the Wound Healing Size Tool (Suarez-Arnedo et al, 2020) in ImageJ/Fiji (Schindelin et al, 2012). The wound area was quantified in the 0 day and 7 day images across 3 replicates per condition. Wound healing rate is shown as the ratio of the scratch area at day 7 vs day 0, with a ratio of 1 representing no migration of cells into the scratch area. Wound Healing Size Tool settings - Variance window radius: 15, Threshold value: 100, Percent saturated pixels 0.1. Sample identity was blinded during analysis.

## Absolute quantification of NADP by LC-MS

Cell harvest and extraction of NADP (including NADP⁺ and NADPH) was based on a previously described method (Lu et al, 2018). Briefly, cells were harvested by washing with PBS twice and quenched on dry-ice by adding acidified 2:2:1 ACN/MeOH/water with spiked-in ¹⁵N₅-ADP (Sigma) as an internal standard. 2 M ammonium bicarbonate was then added to adjust the pH to 8 (22:3 v/v). The extract was freeze-thawed by liquid nitrogen and sonicated three times and centrifuged at 15,000 rpm for 15 min. The supernatant was transferred and subjected to LC-MS analysis. The extract was kept on ice in all steps. LC-MS analysis was performed with the same instrument settings as metabolomics analysis with a different chromatographic method. A Phenomenex Luna-NH2 column (2.1 × 50 mm × 3 μm, 100 Å, Phenomenex) was used for metabolite separation. Mobile phase A consists of 5:95 ACN/water with 20 mM ammonium acetate and 20 mM ammonium hydroxide, and mobile phase B consists of 95:5 ACN/water. The column was operated under room temperature at a flow rate of 0.4 mL/min for analysis and 0.6 mL/min for equilibrium with following gradient: 0–0.5 min 60% B, 3–10 min 0% B, 11.5–15 min

60% B. Linear standard calibration curves were established at a range of 4.9–2500 ng/mL for quantification. The concentrations were normalized by cell numbers for each experimental group.

## Quantification of the pentose phosphate pathway (PPP) flux

Relative PPP Flux was measured by a previously described method using [1,2-$^{13}$C]-glucose (Sigma) as the labeling tracer (Li et al, 2014). Briefly, cells were grown in RPMI media with unlabeled glucose and 10% FBS for 36 h and replenished with RPMI media containing 11.11 mM [1,2-$^{13}$C]-glucose and incubated for 14.5 h. Aliquots of media were taken at the beginning and end of labeling to measure glucose consumption rate. Cells were harvested, extracted, and subjected to LC-MS analysis as described above. Natural abundance correction on the isotopologues of metabolites was performed by using AccuCor software (Su et al, 2017). Relative PPP flux (normalized by glycolysis) was determined with the following equation:

$$Relative\ PPP\ flux = glucose\ consumption\ rate \times \frac{Lac_{M1}}{Lac_{M1} + Lac_{M2}},$$

where $Lac_{M1}$, the M1-labeled lactate, derives from [1,2-$^{13}$C]-glucose that traverses through oxidative PPP and recycled back to glycolysis, and $Lac_{M2}$, the M2-labeled lactate, derives from [1,2-$^{13}$C]-glucose that traverses through glycolysis directly. Relative PPP flux was normalized to the corresponding control group.

## Lipidomics analysis

500 μl of cold methanol was added to cell pellets. After ultrasonication for 30 s, cell lysate was transferred to extraction tubes. 500 μL of $H_2O$ (HPLC grade, Fisher Chemical), 500 μL cold methanol (HPLC grade, Fisher Chemical) and 450 μL DCM (HPLC grade, Fisher Chemical) were added to the cell lysate to form a single phase. Samples were incubated on ice for 30 min. After 30 min, 100 μL Lipidyzer internal standard (SCIEX) was added to the sample mixture, followed by 500 μL of $H_2O$ and 450 μL of DCM. Samples were centrifuged at 1500 rpm for 15 min. Bottom layer was transferred to a new collection tube. 0.9 mL of DCM was then added to the extraction tube for the second extraction. The bottom layer was combined to the same collection tube. Sample extract was then dried under gentle stream of nitrogen and reconstituted in 500 μL buffer (DCM:Methanol (1:1), 10 mM ammonium acetate) for Lipidyzer™ Platform direct infusion analysis (Cao et al, 2020) on AB Sciex 6500 + LC-MS/MS. Flow rate is set at 7 μL/min, injection volume is 50 μL. The autosampler temperature was kept at 15 °C. Buffer A and B are the same as reconstitution buffer (DCM:Methanol (1:1), 10 mM ammonium acetate).

Lipids concentrations were calculated by the Lipidyzer™ platform based on the known concentrations of spiked internal standards. Heatmaps were generated using R.

## Bioinformatics methods

### NRF2 multi-omic feature discovery

To develop a robust multi-omic NRF2 signature, we utilize CRISPR chronos scores and siRNA demeter scores from the DepMap, combining them in an 80:20 weighted average to obtain our predictive vector **y**. We then aggregate associated -omics features (e.g., expression, mutation, copy number, pathways, metabolites, etc.) across pan-cancer cell lines as **X** (depmap.org). To predict **y** from **X**, we utilize the RandomForestRegressor class from sklearn, modifying the default parameters for max_depth=7, min_samples_leaf=2 and setting the number of features per tree (max_features) to 0.5% of the total number of features in **X**, and the number of trees (n_estimators) to 10,000, resulting in 50-fold coverage of each feature in the forest. Training the models using 10-fold cross-validation with an 80:20 train:test split, we subsequently attenuate all tree predictions on **y** proportional to each tree's out-of-bag scores to reduce the weight of poorly predicting trees. Explaining feature importance as impact on model predictions of **y** using SHAP, we rank the features by average SHAP values across all 10 models. NRF2 signature gene were defined as any that met the following criteria: Within the top SHAP features (cumulative fifteen percent) and presence in ≥7 literature datasets, or ≥5 literature sets and a LogFC < 0.25 following 6 h shNRF2 treatment, or ≥2 literature datasets and a LogFC < 0 following 6 h shNRF2 treatment.

### Quantification of signature gene ChIP

ChIP peak values are drawn from public A549 ENCODE datasets for NRF2 (ENCFF414UMG), P300 (ENCFF506BDU), H3K4me3 (ENCFF982HNO), and H3K27ac (ENCFF747IZX). The average value of all peaks within 1 kb up- and down-stream of the promoter are used for the final ChIP promoter score.

To identify the nearest NRF2 peak from each signature gene we used the public A549 ENCODE dataset ENCFF373NIO which is filtered for significance by IDR.

Histone and transcription factor ChIP-seq data are taken from the ENCODE datasets listed in Dataset EV5. Enrichment was quantified by taking all IDR significant NRF2 peaks from ENCFF373NIO that are proximal to a NRF2 signature gene. The locations of these peaks were then intersected with the histone and transcription factor datasets above to find the ChIP signal value of other factors overlapping NRF2 peaks. The mean ChIP signal was calculated for each factor in the low promoter NRF2 and high promoter NRF2 groups. Difference in means was tested for each factor by a Wilcoxon rank-sum test. $P$-values for replicates of the same factor were combined using Fisher's method. To correct for multiple comparisons the $p$-values were adjusted by the Benjamini–Hochberg procedure, and only factors with an adjusted $p$-value < 0.05 were called significant.

### PRO-seq

Exponentially growing A549 and H460 cells harboring doxycycline-inducible shNRF2 (Fig. EV1) were treated with 250 ng/mL dox for 0, 3, or 6 h. For A-485 treatment, A549 shNRF2 cells were treated with 1 μM drug for 1 h. PRO-seq was performed by Arpeggio Biosciences and peaks called via the method detailed in (Mahat et al, 2016). Control peaks were called for bi-directional peaks across 3 replicates for DMSO in both A549 and NCI-H460 cell lines. A-485 (6 h) and shNRF2 time point (0 h, 3 h, and 6 h) peak magnitude was compared against the control peaks and differential peak expression was analyzed using DESeq2. Peak locations were intersected with promoter and enhancer annotations identified by ChromHMM. PRO-seq peaks were intersected with the same ENCODE datasets in *Quantification of NRF2 signature gene ChIP* to obtain NRF2 and P300 ChIP at regions overlapping PRO-seq

peaks. Downstream analysis of PRO-seq peak links by H3K27ac HiChIP simulation was limited to peaks with a log fold change < 0 and a $p$-value < 0.1 in at least 3 out of the 4 possible conditions: 3 h A549 and 6 h A549 for a final list of consensus peaks and their associated A549 ChIP values.

### H3K27ac ChIP-seq processing

H3K27ac FASTQ files were downloaded from NCBI GEO (Table EV2). FASTQ files were processed via the ENCODE Histone ChIP-seq pipeline for replicated experiments (https://github.com/ENCODE-DCC/chip-seq-pipeline2). In short: FASTQ reads were mapped to GRCh38 using Bowtie2 followed by post-alignment filtering of low quality, unmapped (samtools view -F 1804 -f 2 -q 30) and duplicate (picard MarkDuplicates) reads. Peaks were then called using MACS2 (https://www.biorxiv.org/content/10.1101/496521v1, PREPRINT) using DNA input only controls.

Differential binding of H3K27ac following siNRF2 treatment was quantified using DiffBind (http://bioconductor.org/packages/release/bioc/vignettes/DiffBind/inst/doc/DiffBind.pdf) to find consensus peaks across samples and quantify peak false discovery rate and log2 fold change between control and siRNA conditions. H3K27ac signal across NRF2-dependent and -independent lines was quantified by taking the average signal fold change vs an input only control over the genomic regions identified by PRO-seq.

$P$-values for panel 2E are derived from a linear mixed-effects model, with NRF2-dependency (Yes/No) and responsive peak class (Promoter/Enhancer/Linked) as fixed effects and cell line and parent project as random effects, followed by multiple testing correction via Tukey's method.

### H3K27ac HiChIP simulation from Hi-C and ChIP data

Simulated H3K27ac HiChIP was generated using FitHiChIP, which simulates HiChIP contact maps given a HiC contact map and ChIP-seq data (Bhattacharyya et al, 2019). ENCODE datasets for H3K27ac ChIP (ENCFF508YMR) and A549 HiC (ENCFF121YPY) were processed using FitHiChIPs HiC_Simulate_by_ChIP_Coverage. R script using a 5000 nucleotide resolution.

Linked regions were filtered to have a FitHiChIP generated contact count score (CCscore) of at least 1. Downstream analysis was limited to linked regions where both linkage arms overlapped with a shNRF2 responsive PRO-seq peak (as defined under *Pro-seq*) using the FindOverlaps function from the GenomicRanges R package (Lawrence et al, 2013) (https://bioconductor.org/packages/release/bioc/html/GenomicRanges.html).

To increase confidence in the simulation, we shuffled the dataset 1000 times using bedtools shuffle for BEDPE files [https://bedtools.readthedocs.io/en/latest/]. Chromosome start and end locations in the shuffled data were aligned to the original by flooring each to the nearest 5 kb, matching the resolution and binning of the original dataset. Each shuffled dataset was then processed identically to the original dataset. A simulated link was defined as being present in the shuffled dataset if it was present with a CCscore equal to or greater than the original CCscore. Only links that appeared in less than 5% of the shuffled datasets were deemed significant and analyzed further.

We then further filtered the significant links for only those where at least one of the linkage arms was proximal to a NRF2 signature gene. This final list of linkages was used to quantify the ChIP levels at distal PRO-seq peaks linked to NRF2 signature genes.

### Genome track figures

Genome track figures were generated using pyGenomeTracks (Lopez-Delisle et al, 2021) [https://pygenometracks.readthedocs.io/en/latest/]. Simulated linkages are colored by significance (as described under *H3K27ac HiChIP*).ChIP tracks are from ENCODE and represent the signal fold change for NRF2 (ENCFF796HRU), P300 (ENCFF052CLF), H3K4me3 (ENCFF152LRB), and H3K27ac (ENCFF105IIK). All NaN values for any ChIP track were set to 0. Only RefSeq promoter annotations that are within a shNRF2 responsive PRO-seq peak or are the annotated promoter of a NRF2 signature gene are shown. H3K27ac changes in response to siNRF2 are plotted as in the original Okazaki 2020 dataset (Okazaki et al, 2020) except mapped to GRCh38. In short, peaks were called by MACS2 and resulting treat_pileup.bdg files were used as occupancy profiles. Bedgraph files were first scaled to the total number of mapped reads using wiggletools scale and then individual wig files were combined using wiggletools mean (Zerbino et al, 2014). Gene annotations are based on canonical transcripts from GENCODE 27.

### Bulk RNA-sequencing

RNA-sequencing data were analyzed using HTSeqGenie (under software) in Bioconductor (Huber et al, 2015) as follows: first, reads with low nucleotide qualities (70% of bases with quality <23) or matches to rRNA and adapter sequences were removed. The remaining reads were aligned to the human reference genome (human: GRCh38.p10) using GSNAP (Wu and Nacu, 2010; Wu et al, 2016) version '2013-10-10-v2', allowing maximum of two mismatches per 75 base sequence (parameters: '-M 2 -n 10 -B 2 -i 1 -N 1 -w 200000 -E 1 --pairmax-rna=200000 --clip-overlap'). Transcript annotation was based on the GENCODE genes database (human: GENCODE 27). To quantify gene expression levels, the number of reads mapping unambiguously to the exons of each gene was calculated. Differential expression analysis was performed with limma-voom (Law et al, 2014).

### Exogenous NRF2 re-expression

Genes were required to have an average expression (in $\log_2$ CPM) of −1 for downstream analysis. To account for effects of the re-expression system, we normalized $\log_2 fc$ values for wildtype NRF2 (24 vs 0 h) and Neh4/5mut (24 vs 0 h) by subtracting the $\log_2 fc$ values for Luciferase (24 vs 0 h). Negative values represent a decrease in expression compared to Luciferase and positive values represent an increase. Variability between genes with increased expression in both Neh4/5mut and wildtype NRF2 re-expression was quantified by the cosine similarity of each point and its closest point on the diagonal line representing a 1:1 expression between the two conditions. Cosine similarity was calculated using the Cosine Measure function in Latent Semantic Analysis R package (under software).

## Data availability

RNA-/PRO-seq datasets have been deposited to GEO as GSE282090, GSE282091, GSE282300. Nuclear interactomics raw

data have been deposited to MassIVE with idenfitier MSV000096356. Metabolomics and lipidomics data have been deposited to the Metabolomics Workbench (Sud et al, 2016) as project PR002208.

The source data of this paper are collected in the following database record: biostudies:S-SCDT-10_1038-S44319-025-00463-z.

## Peer review information

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

## Acknowledgements

We gratefully acknowledge support from Grace Chan, Jessica Grandner, Madeleine Prangley, Tommy Cheung, Anita Izrael-Tomasevic, David Stokoe, Joey Azofeifa, Christiaan Klijn, David Arnott, Kebing Yu, Rajini Srinivasan, Karen Gascoigne, and Robert Yauch during the course of this work. This work was funded by Genentech/Roche. Synopsis figure was generated using BioRender.

## Author contributions

**Ryan J Conrad**: Conceptualization; Data curation; Formal analysis; Writing—original draft; Writing—review and editing. **James A Mondo**: Data curation; Software; Formal analysis; Writing—review and editing. **Mike Lingjue Wang**: Data curation; Formal analysis. **Peter S Liu**: Data curation; Formal analysis. **Zijuan Lai**: Data curation; Formal analysis; Supervision. **Feroza K Choudhury**: Data curation; Formal analysis; Writing—review and editing. **Qingling Li**: Data curation; Formal analysis. **Weng Ruh Wong**: Data curation. **James Lee**: Data curation. **Frances Shanahan**: Data curation. **Eva Lin**: Data curation. **Scott Martin**: Formal analysis; Supervision. **Joachim Rudolph**: Supervision. **John G Moffat**: Supervision. **Dewakar Sangaraju**: Formal analysis; Supervision. **Wendy Sandoval**: Formal analysis; Supervision. **Timothy Sterne-Weiler**: Formal analysis; Supervision; Methodology; Writing—review and editing. **Scott A Foster**: Conceptualization; Supervision; Writing—original draft; Writing—review and editing.

Source data underlying figure panels in this paper may have individual authorship assigned. Where available, figure panel/source data authorship is listed in the following database record: biostudies:S-SCDT-10_1038-S44319-025-00463-z.

## Disclosure and competing interests statement

All authors of the paper are or were employees/shareholders of Genentech/Roche during the execution of this work.

# Expanded View Figures

**Figure EV1.  Characterization of an optimized NRF2 transcriptional signature.**

(**A**) Comparison of expression features (red) predicting NRF2 dependency from the multi-omic feature analysis compared to NRF2 target genes identified in the literature (gray). Left - Proportional overlap of expression features with literature datasets. Number of genes identified in each literature set noted above the bars. Right - Proportion of genes that are identified in multiple literature sets that are also identified as top expression features. (**B**) Number of literature references for each NRF2 signature gene. (**C**) Western blot of shNRF2 or shNTC A549 and NCI-H460 cells +/−400 ng/mL dox. (**D**) Incucyte growth curves of shNTC or shNRF2 A549 or NCI-H460 cells with indicated dox concentrations. Validation data are from one experiment, with mean and SEM derived from 16 images/well indicated. (**E**) Volcano plots of RNA-seq values of NRF2 signature genes (pink) and compared to non-signature other genes (gray) in A549 and NCI-H460 cells shNRF2 cells treated with 500 ng/mL dox for indicated time points. Data are expressed relative to pre-treatment control Time points are relative to pre-treatment (0 h). Data are derived from two independent samples. (**F**) Distribution and density of gene counts across promoter NRF2 ChIP signal levels. The histogram represents the number of genes at each ChIP signal. The peak (dashed line) of the estimated probability density function (gray region) serves as the cut-off point for classification into group I (low promoter NRF2, green) and group II (high promoter NRF2, purple).

▶

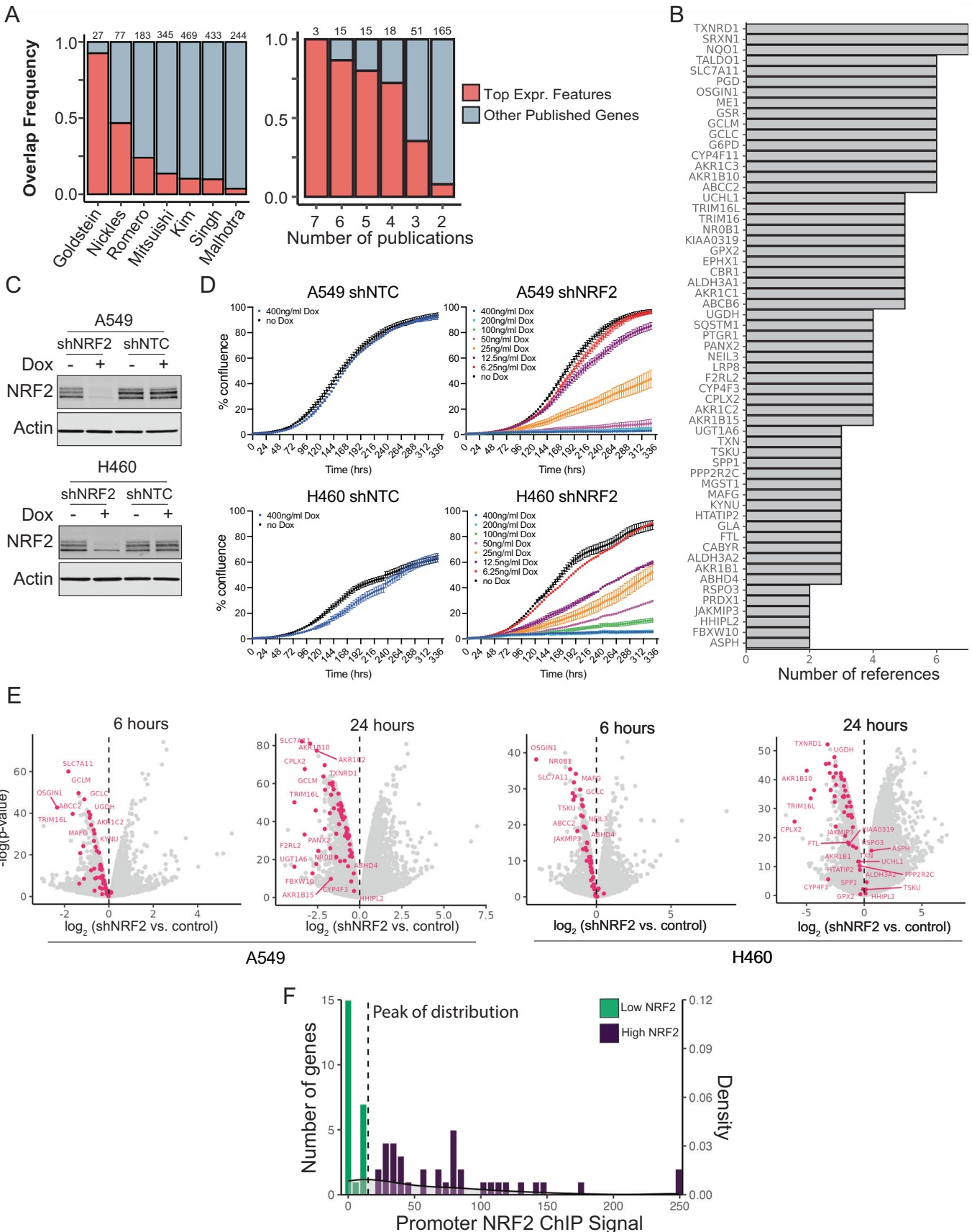

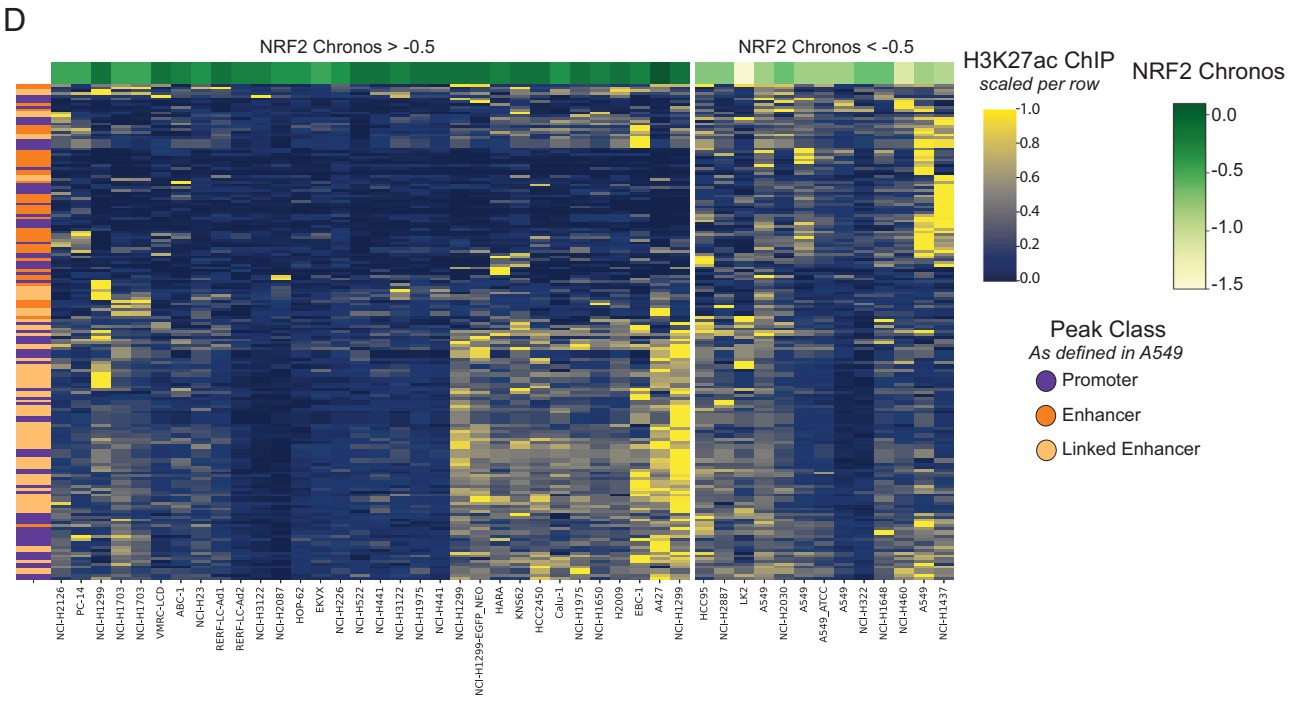

**Figure EV2.   NRF2 is required for eRNA transcription and H3K27ac deposition.**

(**A**) Volcano plot of the log$_2$ fold change of the sum of all PRO-seq reads within a gene region. A549 and H460 shNRF2 treatment is 250 ng/mL dox at 3 and 6 h relative to control (water). Data are derived from three replicates for A549, and from two individual replicates for H460. (**B**) Volcano plot of individual (gene-agnostic) PRO-seq peaks in A549 shNRF2 at 3 h vs control (water). Data are derived from three replicates for A549, and from two individual replicates for H460. (**C**) Volcano plot of individual (gene-agnostic) PRO-seq peaks in H460 shNRF2 at 6 h vs control (water). Data are derived from three replicates for A549, and from two individual replicates for H460. (**D**) Heatmap of H3K27ac ChIP (fold change compared to input control) with 1 representing the highest value for every row. Rows represent genomic regions containing NRF2 responsive PRO-seq peaks and are colored by classification.

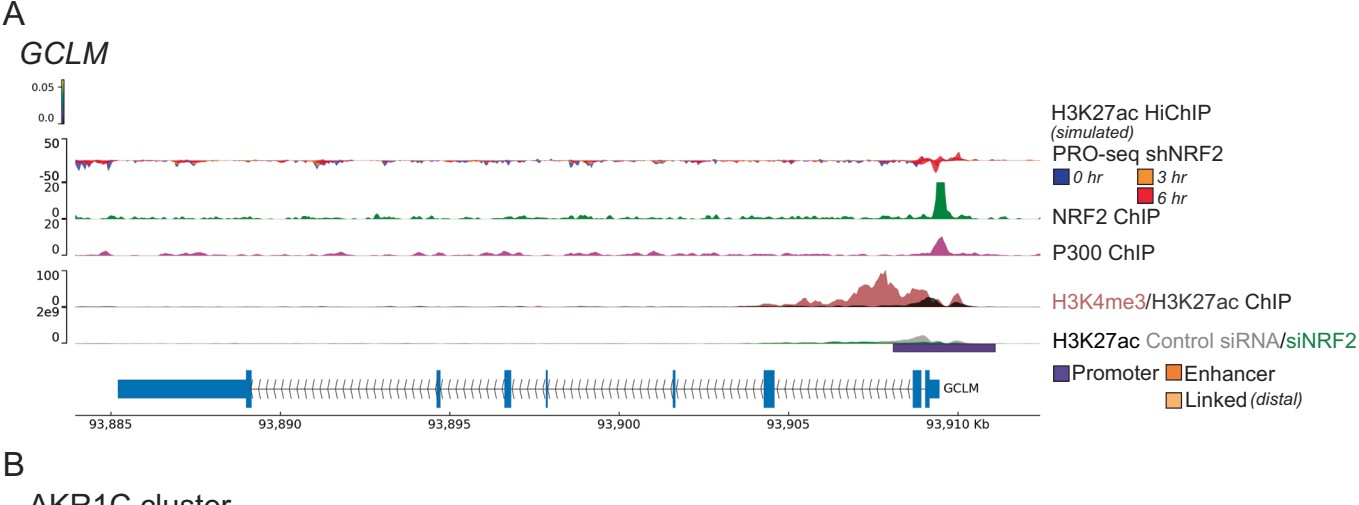

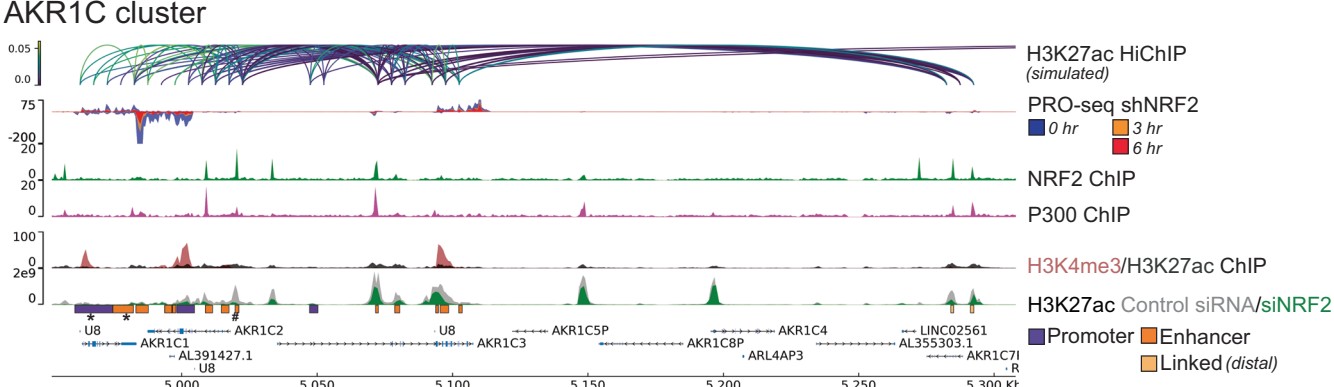

**Figure EV3. Illustrative genome browser snapshots of group I and group II signature genes.**

(A) Genome browser snapshots of the group II NRF2 signature gene GCLM highlighting the neighboring chromatin landscape. Simulated H3K27ac HiChIP links are colored by *p*-value. (B) Genome browser snapshots of the group I NRF2 signature genes AKR1C1-3 highlighting the highly linking chromatin landscape of this gene cluster. Simulated H3K27ac HiChIP links are colored by *p*-value. * denotes PRO-seq peaks from Fig. 2A. # denotes H3K27ac peak that is NRF2 responsive from Fig. 2D.

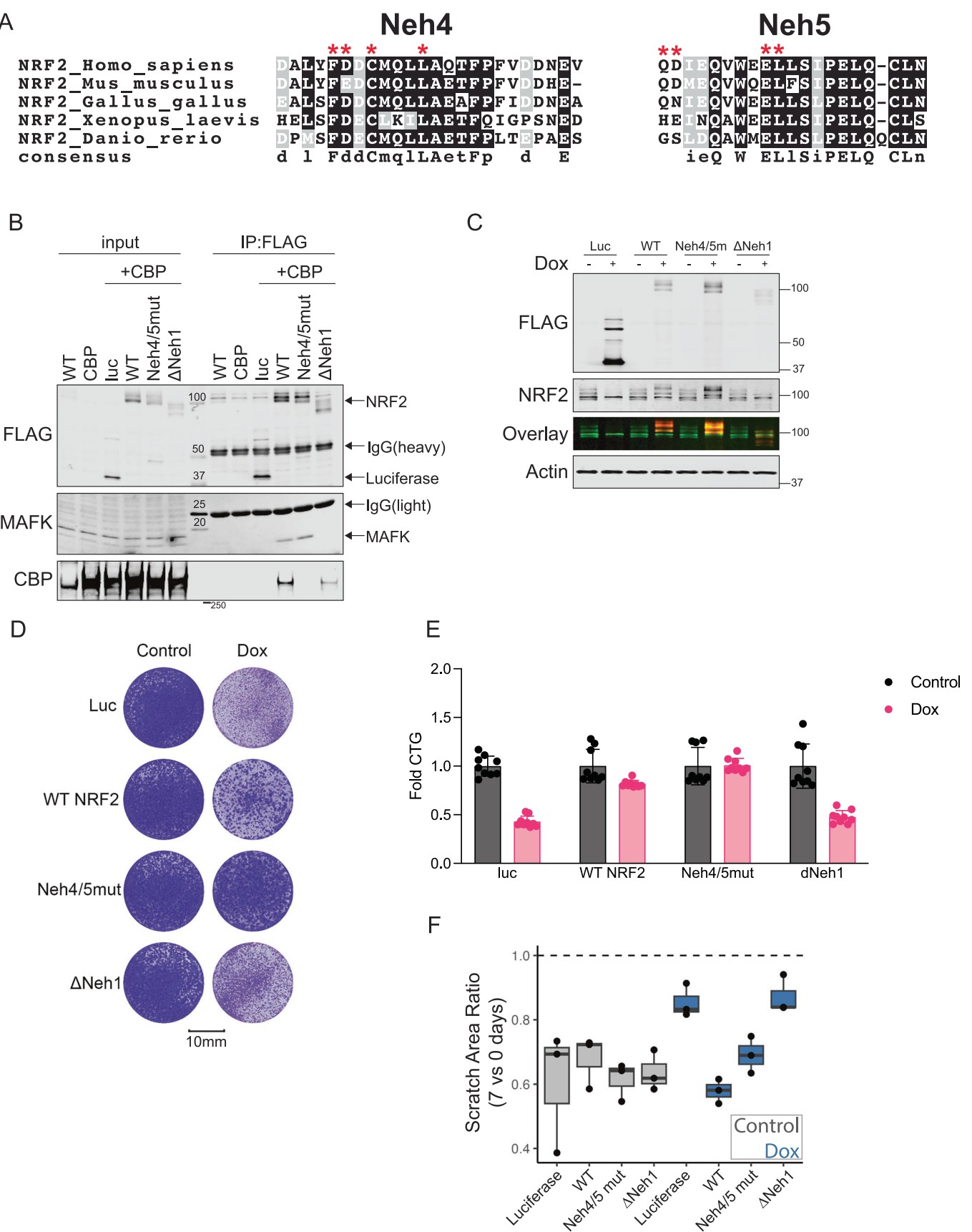

**Figure EV4. Definition and validation of a CBP/p300-deficient NRF2, Neh4/5mut.**

(A) Alignment of metazoan NRF2 Neh4 and Neh5 segments with Neh4/5mut substitutions indicated. Mutation of QD in Neh5 was derived from (Zhang et al, 2007). (B) Western blot of input (left) and FLAG-immunoprecipitations (right) of FLAG-tagged WT, Neh4/5mut, ΔNeh1, or luciferase control expressing HEK293 cells co-transfected with or without GFP-tagged CBP. Data are representative of three independent experiments. (C) Western blotting of H460-shNRF2-BIND-luc, -BIND-WT NRF2, -BIND-Neh4/5mut, or -BIND-ΔNeh1 cells treated with water (control) or 2.5 ng/mL dox for 24 h probing with FLAG, NRF2, or β-actin antibodies. Representative data from three independent experiments is shown. (D) Crystal violet staining of H460-shNRF2-BIND-luc, -BIND-WT NRF2, -BIND-Neh4/5mut, or -BIND-ΔNeh1 treated with water (control) or 2.5 ng/mL dox for 8 days. Representative data from three independent experiments is shown. (E) CellTiter-Glo measurements of H460-shNRF2-BIND-luc, -BIND-WT NRF2, -BIND-Neh4/5mut, or -BIND-ΔNeh1 treated with water (control) or 2.5 ng/mL dox for 5 days. Data are expressed as a fold of control values, with SD indicated and are derived from three individual experiments. (F) Quantification of scratch assay wound healing in A549shNRF2-BIND cells +/−2.5 ng/mL dox imaged over 7 days on the Incucyte system. Wound healing rate is shown as the ratio of the scratch area at day 7 vs day 0, with a ratio of 1 representing no migration of cells into the scratch area. Data are derived from three individual replicates.

A

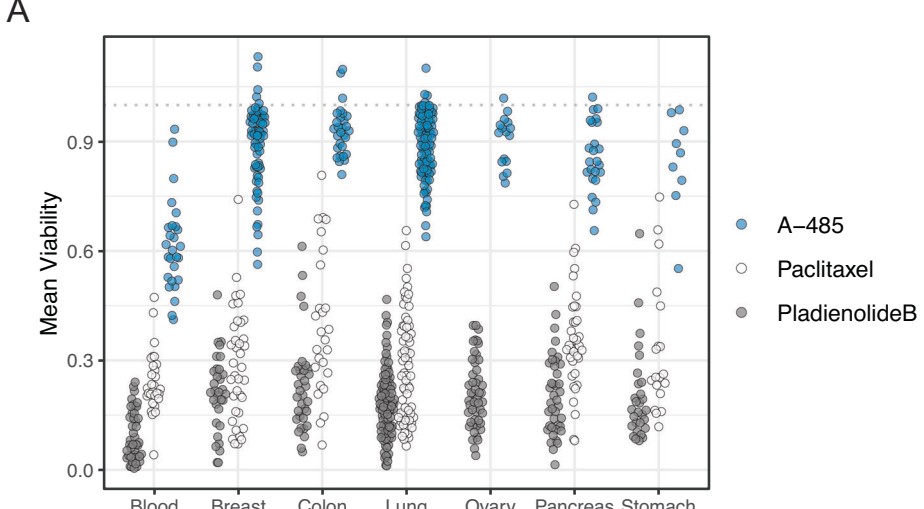

B

**Figure EV5.  Cell viability responses upon A-485 treatment across diverse cancer cell lines.**

(A) Mean viability across 535 cell lines treated with a dose response of A-485 or the broadly active compounds (Paclitaxel or Pladienolide B) in 5-day viability assays determined by CellTiter-Glo. (B) Crystal violet staining of 14 cell lines, eight NRF2-dependent (Chronos < −0.5) and 6 NRF2-independent (Chronos > −0.5) treated with indicated concentrations of A-485 or A-486 control for 7 days. Data are representative of 2 independent experiments.

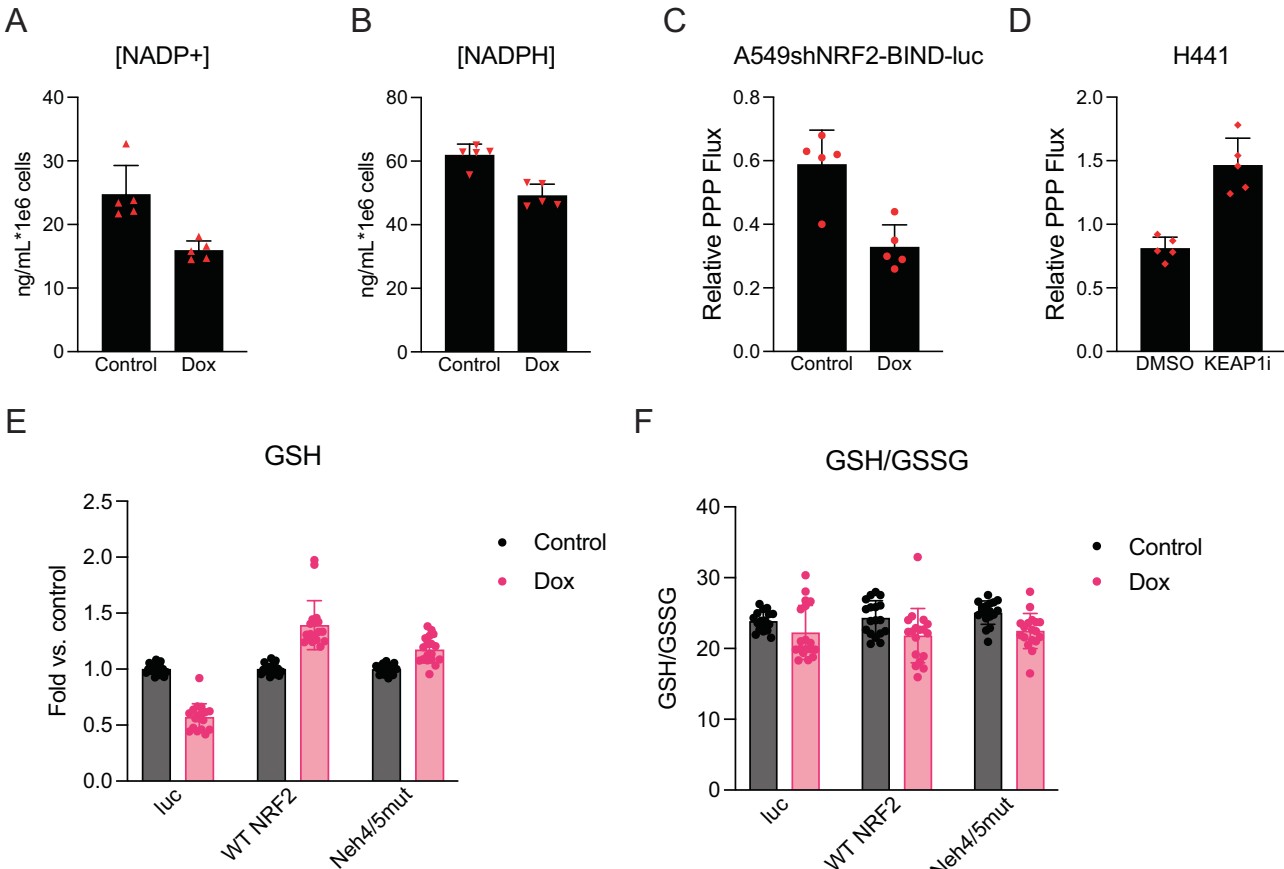

**Figure EV6. Metabolic characterization of A549shNRF2-/H460shNRF2-BIND rescue cell models.**

Mean NADP+ (**A**) and NADPH (**B**) levels in A549shNRF2-BIND-luc cells treated with water (Control) or 2.5 ng/mL doxycycline (Dox) for 48 h. Data are derived from 5 individual replicates with SD indicated. (**C**) Mean relative PPP Flux in A549shNRF2-BIND-luc cells treated with water (Control) or 2.5 ng/mL doxycycline (Dox) for 48 h and labeled with [1,2-$^{13}$C]-glucose for 14.5 h. Data are derived from 5 individual replicates with SD indicated. (**D**) Mean relative PPP Flux in H441 cells treated with DMSO control or 1 µM KEAPi (Davies et al, 2016) for 48 h and labeled with [1,2-$^{13}$C]-glucose for 14.5 h. Data are derived from 5 individual replicates with SD indicated. (**E**) Mean GSH levels in A549shNRF2-BIND-luc, -BIND-WT NRF2, -BIND-Neh4/5mut cells treated with water (control) or 2.5 ng/mL dox for 48 h. 2. Data represent three independent experiments with SD indicated. (**F**) Mean GSH/GSSG ratios in A549shNRF2-BIND-luc, -BIND-WT NRF2, -BIND-Neh4/5mut cells treated with water (control) or 2.5 ng/mL dox for 48 h. rescued with luciferase control, WT or Neh4/5mut NRF2. Data represent three independent experiments with SD indicated.

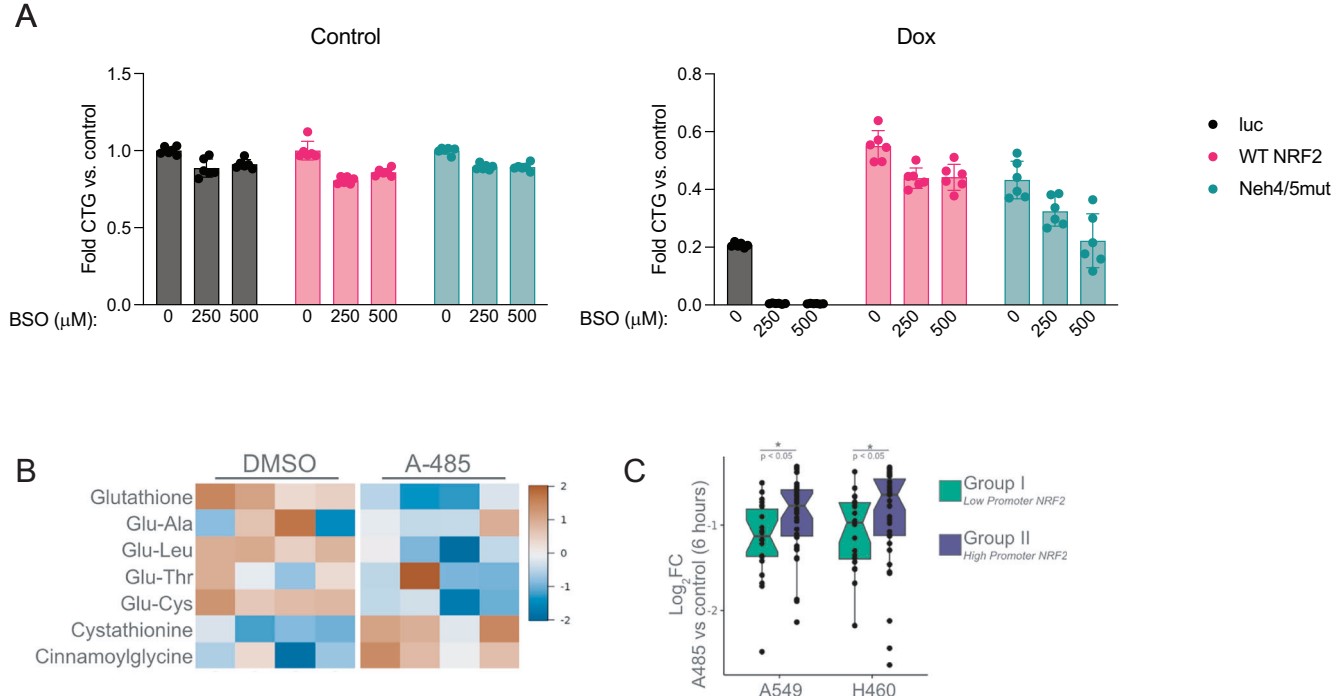

**Figure EV7. BSO sensitivity in the A549shNRF2-BIND rescue system and mechanistic characterization of A-485 treatment in NRF2-dependent NSCLC models.**

(A) Mean CellTiter-Glo measurements of A549shNRF2-BIND-luc, -BIND-WT NRF2, -BIND-Neh4/5mut cells treated with water (control) or 2.5 ng/mL dox for 48 h, then treated with indicated concentrations of BSO for 96 h. Data are expressed as a fold relative to parental control (no dox, no BSO), and are derived from two individual replicates performed in triplicate with SD indicated. (B) Heatmap of GSH pathway metabolites intensity normalized by row as a z score from A549shNRF2-BIND-luc cells treated with 1 μM A-485 for 24 h. Each column represents an individual replicate. (C) RNA-seq values of group I vs group II (Fig. 1B) genes from A549 and H460 cells treated with 1 μM A-485 for 6 h. Values show limma-voom $\log_2$ fold changes between A-485 treatment vs DMSO. Data are derived from three individual replicates. *P*-values derived from a Wilcoxon rank-sum test, followed by adjustment by the Benjamini–Hochberg procedure to correct for multiple testing. Box represents the first and third quartiles, with the median shown as a horizontal line. Whiskers extend to the smallest and largest value within 1.5 * the interquartile range. Notches extend 1.58 *IQR/ sqrt(*n*). Group I $n = 23$, group II $n = 36$.

