## [Peer Review File · EMBO Reports]

NRF2 supports non-small cell lung cancer growth independent of CBP/p300-enhanced glutathione synthesis

Scott Foster, Ryan Conrad, James Mondo, Mike Lingjue Wang, Peter Liu, Zijuan Lai, Feroza Choudhury, Qingling Li, Weng Ruh Wong, James Lee, Frances Shanahan, Eva Lin, Scott Martin, Joachim Rudolph, John Moffat, Dewakar Sangaraju, Wendy Sandoval, and Timothy Sterne-Weiler

Corresponding author(s): Scott Foster (foster.scott@gene.com) , Timothy Sterne-Weiler (sternewt@gene.com)

Review Timeline:

Submission Date:	2nd Apr 24
Editorial Decision:	3rd Jun 24
Revision Received:	5th Nov 24
Editorial Decision:	17th Feb 25
Revision Received:	10th Mar 25
Accepted:	7th Apr 25

Transaction Report:

Dear Dr. Foster

Thank you for the submission of your research manuscript to our journal. I apologize for the delay in handling your manuscript. It has been sent to 4 experts and we have meanwhile received the reports from 3 of them (copied below). The 4th report is expected to be delivered within the next few days and if so, I will forward it to you.

As you will see, the referees acknowledge that the findings are interesting and of value to the field, but they also raise a number of concerns and point out that further validation will be required. I feel that all points are pertinent and would need to be addressed in the revision.

Given the constructive comments, we would like to invite you to revise your manuscript with the understanding that the referee concerns (as detailed above and in their reports) must be fully addressed and their suggestions taken on board. Please address all referee concerns in a complete point-by-point response. Acceptance of the manuscript will depend on a positive outcome of a second round of review. It is EMBO Reports policy to allow a single round of revision only and acceptance or rejection of the manuscript will therefore depend on the completeness of your responses included in the next, final version of the manuscript.

We realize that it is difficult to revise to a specific deadline. In the interest of protecting the conceptual advance provided by the work, we recommend a revision within 3 months (September 3rd). Please discuss the revision progress ahead of this time with the editor if you require more time to complete the revisions.

I am also happy to discuss the revision further via e-mail or a video call, if you wish.

A few general comments:

- Your manuscript contains a number of 'omics' data, please deposit these datasets in a public repository before submission of the revised manuscript (see point 7 below).
- Please also note point 11 below - the use of data citations to acknowledge not only the manuscript reporting e.g. ChIP-seq data but also the dataset deposited on e.g. ENCODE.
- Complex tables, such as e.g. Table EV4 should be uploaded as Datasets (Dataset EV#) with a legend in a separate tab.
- Finally, we encourage depositing Code and Software on Github and to add the respective links to the Data Availability section.

2) individual production quality figure files as .eps, .tif, .jpg (one file per figure).

Please download our Figure Preparation Guidelines (figure preparation pdf) from our Author Guidelines pages <https://www.embopress.org/page/journal/14693178/authorguide> for more info on how to prepare your figures.

4) a complete author checklist, which you can download from our author guidelines (<<https://www.embopress.org/page/journal/14693178/authorguide>>). Please insert information in the checklist that is also reflected in the manuscript. The completed author checklist will also be part of the RPF.

5) Please note that all corresponding authors are required to supply an ORCID ID for their name upon submission of a revised manuscript (<<https://orcid.org/>>). Please find instructions on how to link your ORCID ID to your account in our manuscript tracking system in our Author guidelines (<<https://www.embopress.org/page/journal/14693178/authorguide#authorshipguidelines>>)

6) We replaced Supplementary Information with Expanded View (EV) Figures and Tables that are collapsible/expandable online. A maximum of 5 EV Figures can be typeset. EV Figures should be cited as 'Figure EV1, Figure EV2' etc... in the text and their respective legends should be included in the main text after the legends of regular figures.

7) Before submitting your revision, primary datasets (and computer code, where appropriate) produced in this study need to be deposited in an appropriate public database (see < <https://www.embopress.org/page/journal/14693178/authorguide#dataavailability>>).

The accession numbers and database should be listed in a formal "Data Availability" section (placed after Materials & Method) that follows the model below (see also < <https://www.embopress.org/page/journal/14693178/authorguide#dataavailability>>). Please note that the Data Availability Section is restricted to new primary data that are part of this study.

Data availability

Additional information on source data and instruction on how to label the files are available <<https://www.embopress.org/page/journal/14693178/authorguide#sourcedata>>.

10) Figure legends and data quantification:
The following points must be specified in each figure legend:

- the name of the statistical test used to generate error bars and P values,
- the number (n) of independent experiments (please specify technical or biological replicates) underlying each data point,
- the nature of the bars and error bars (s.d., s.e.m.)

- If the data are obtained from n {less than or equal to} 5, show the individual data points in addition to the SD or SEM.
- If the data are obtained from n {less than or equal to} 2, use scatter blots showing the individual data points.

See also the guidelines for figure legend preparation:
<https://www.embopress.org/page/journal/14693178/authorguide#figureformat>

11) Our journal encourages inclusion of *data citations in the reference list* to directly cite datasets that were re-used and obtained from public databases. Data citations in the article text are distinct from normal bibliographical citations and should directly link to the database records from which the data can be accessed. In the main text, data citations are formatted as follows: "Data ref: Smith et al, 2001" or "Data ref: NCBI Sequence Read Archive PRJNA342805, 2017". In the Reference list, data citations must be labeled with "[DATASET]". A data reference must provide the database name, accession number/identifiers and a resolvable link to the landing page from which the data can be accessed at the end of the reference. Further instructions are available at <<https://www.embopress.org/page/journal/14693178/authorguide#referencesformat>>.

12) All Materials and Methods need to be described in the main text. We would encourage you to use 'Structured Methods', our new Methods format. According to this format, the Methods section should include a Reagents and Tools Table (listing key

reagents, experimental models, software and relevant equipment and including their sources and relevant identifiers) followed by a Methods and Protocols section in which we encourage the authors to describe their methods using a step-by-step protocol format with bullet points, to facilitate the adoption of the methodologies across labs. More information on how to adhere to this format as well as downloadable templates (.doc or .xls) for the Reagents and Tools Table can be found in our author guidelines: < <https://www.embopress.org/page/journal/14693178/authorguide#manuscriptpreparation>>.

An example of a Method paper with Structured Methods can be found here:
<<https://www.embopress.org/doi/10.15252/msb.20178071>>.

13) As part of the EMBO publication's Transparent Editorial Process, EMBO Reports publishes online a Review Process File to accompany accepted manuscripts. This File will be published in conjunction with your paper and will include the referee reports, your point-by-point response and all pertinent correspondence relating to the manuscript.

Kind regards,

Referee #1:

In their investigation, Conrad and colleagues delve into the intricate role of NRF2, a transcription factor often implicated in the progression of aggressive lung cancers, shedding light on its influence on cancer cell proliferation. Their study uncovers a fresh array of genes under NRF2 regulation, unveiling a significant enhancer activity linked to NRF2. Intriguingly, they challenge prior assumptions by revealing that the NRF2-CBP/p300 interaction, long thought crucial for cancer cell growth, may not be as pivotal as once believed. Metabolic scrutiny suggests that while CBP/p300 contributes notably to glutathione production, essential for redox homeostasis, NRF2-dependent cancer growth appears to operate independently of this pathway.

While valuable, the study warrants further scrutiny. Some assertions may benefit from moderating, a point the authors may want to consider.

Exploring the NRF2 dependency could provide valuable insights. Have the authors attempted to rescue the growth deficiencies of NRF2-silenced cells by supplementing with NAC, bME, or other antioxidants? The study by Vartanian et al. (2019), referenced in the discussion, only explores GSH, yet this is problematic due to inefficient GSH uptake.

The experimental design predominantly relies on a somewhat intricate system where an inducible shRNA downregulates NRF2 while simultaneously inducing the expression of luciferase, WT-NRF2, and various mutants. This complexity may raise questions regarding whether genotype-specific differences stem solely from differing kinetics of expression and downregulation. In Figure 3E, unless I misunderstood something, why the NRF2 ab is unable to recognize the Flag-constructs?

Why not validate some findings in an NRF2-knockout cell line reconstituted with the inducible system? In our group's experience, A549 cells can be readily engineered to lack NRF2. Establishing such a system would allow for proper quantification of GSH, an aspect somehow unclear. Figure EV6 presents data only in fold versus control, hindering comparisons between genotypes.

The study would benefit by showing that WT and Neh4 mutants have a similar transactivation activity in an ARE-Luc assays which is routinely employed to address NRF2 activity.

Exploring the sensitivity of NRF2-deficient, WT-NRF2, and Neh4 mutants to GSH-depleting agents such as BSO and inhibitors

of TXNRD1 (e.g., TRi-1) could also provide important insights into the observations reported here.

The CREBBP/EP300 double KO cells are not satisfactorily characterized. Do they also show decreased GSH levels? Do they show loss of expression of group1 NRF2 target genes?

Referee #3:

In their study "NRF2 drives non-small cell lung cancer growth independently of CBP/p300-enhanced glutathione synthesis" the authors use several omic-based screens, genomic and bioinformatic analyses to evaluate the role of NRF-2 in lung cancer cells. This data-reach study addresses an important aspect of cancer cell biology i.e. the involvement of redox regulation. As such, I am convinced that the datasets provided here might be of use for researchers working in the field.

Comments and suggestions:

- A more detailed yet understandable explanation of how the 59 NRF2-controlled target genes were selected would be helpful.
- All immunoblots need quantification and statistical evaluation, i.e., they should be repeated multiple times.
- The cell biology-based evaluation of the omic/genomic and bioinformatic results should be more robust to support the study's claims (see title and abstract).
- Using only one technique to evaluate cell viability is insufficient. Cell titer glow is a detection system that depends on cell metabolic state, which might be problematic in a situation where cell metabolism is affected (following NRF2 down-regulation for example)
- Additional measurements of cancer cell biology, such as migration and invasion assays, would help in understanding the contribution of the findings to cancer cell biology.
- Detecting GSH and H₂O₂ using commercial kits known to be prone to artifacts is risky and may affect the conclusions. More advanced techniques, such as EPR or genetically encoded sensors, should be considered for detecting H₂O₂ and GSH/GSSG.
- A second cancer cell entity should be used for the most relevant experiments. For example, melanoma cells seem to be highly reliable on redox regulation.
- Using inhibitors of GSH synthesis, such as BSO or Auranofin, as well as H₂O₂ scavengers, could help verify the authors' statements regarding the role of these two metabolites in cancer cell biology.

Referee #4:

The manuscript by Conrad et al. provides a very innovative and elegant piece of work in which the authors robustly demonstrate that Nrf2 binding to its co-activator CBP/p300 is required for transcription of a set of genes involved in GSH-de novo synthesis but not in cancer cell proliferation. The study is extremely well done. Only a few concerns need to be addressed.

The main one is: "How broad and comprehensive is the message the authors want to convey?"

Looking at the results, it seems that the only Nrf2 ablation (with any external oxidative insult) leads to a significant decrease of GSH (and glutamate-containing dipeptides) and an increase in H₂O₂ levels. This, however, has no effect on cell proliferation, as proven by the authors. At the same time, conditions of Nrf2 knockdown lead to a decrease of one glycolytic metabolite (i.e., F1,6-BP) and an increase of sedoheptulose-7-P, which is a marker of PPP. By comprehensively looking at these results, I've gotten the impression that Nrf2-downregulation leads to general metabolic reprogramming. Specifically, it seems that Nrf2-KO cells redirect glucose to the PPP for the generation of NADPH and provide reducing equivalents to the antioxidant system, which is not "transcriptionally sustained" by Nrf2. This makes glutamine the principal metabolic source for obtaining ATP from the mitochondria (hence the increase of H₂O₂). The decrease in nucleotide levels further corroborates the idea that glucose undergoes oxidative decarboxylation primarily to produce NADPH via the oxidative branch of the PPP without being converted to ribose or giving rise to IMP or CMP.

An explanation of all this rewiring is that the effects the authors are seeing might be due to clonal selection. How many clones did they screen? The authors should at least perform other analyses in a cell system known to be not reliant on glutamine or (alternatively) perform additional experiments with different A-549 clones grown in media that are not supplemented with glutamine, glucose, or inhibited to use lipids to fuel mitochondrial metabolism. This consideration is even more relevant, given the altered lipid composition the authors observe in this condition.

Investigating whether these cells (more clones) are more vulnerable to H₂O₂-mediated insults or whether they have evolved to withstand oxidative stress is another important topic that deserves to be considered.

Minor points:

- The authors should add more details in the figure legends. They are too short and provide no information or clues to understand the figures.
- The authors should provide molecular weights on the western blots. Especially for Nrf2 (whose molecular weight is always debated), this information is fairly relevant.
- Besides GSH values, the authors should also provide the extent of GSSG. This value is important to understand whether the increase in GSH observed upon re-expression of WT Nrf2 is due (mainly or uniquely) due to an increase in the rate of de novo synthesis or reduction. In the light of what discussed above, defining this issue is very relevant.

Response to Reviewer

Figure Legends

Referee #1:

In their investigation, Conrad and colleagues delve into the intricate role of NRF2, a transcription factor often implicated in the progression of aggressive lung cancers, shedding light on its influence on cancer cell proliferation. Their study uncovers a fresh array of genes under NRF2 regulation, unveiling a significant enhancer activity linked to NRF2. Intriguingly, they challenge prior assumptions by revealing that the NRF2-CBP/p300 interaction, long thought crucial for cancer cell growth, may not be as pivotal as once believed. Metabolic scrutiny suggests that while CBP/p300 contributes notably to glutathione production, essential for redox homeostasis, NRF2-dependent cancer growth appears to operate independently of this pathway.

While valuable, the study warrants further scrutiny. Some assertions may benefit from moderating, a point the authors may want to consider.

We thank the referee for carefully reviewing our manuscript and seeing value in the work presented. We have made text edits to moderate some of our claims (i.e., supports vs. drives in the Title, specifying work as derived from cell lines in the Abstract and Introduction, in addition to other language in the text) .

Exploring the NRF2 dependency could provide valuable insights. Have the authors attempted to rescue the growth deficiencies of NRF2-silenced cells by supplementing with NAC, bME, or other antioxidants? The study by Vartanian et al. (2019), referenced in the discussion, only explores GSH, yet this is problematic due to inefficient GSH uptake.

We agree that GSH permeability presents an important caveat in the interpretation of rescue data cited, however, as shown in Figure 2D in Vartanian et al. (2019), addition of GSH does suppress ROS levels under siNRF2 conditions as well as upon TBHP treatment. This indicates some extent of GSH entry into the cell and suppression of the elevated ROS levels upon NRF2 knockdown or TBHP treatment. Additionally, in Supplementary Fig. S6D Vartanian et al. (2019) show that NAC is insufficient to rescue the growth defect. We have performed NAC experiments in our A549shNRF2-BIND system and observe that 2mM NAC did not rescue NRF2 knockdown, similar to these published results.

CellTiter-Glo measurements of A549shNRF2-BIND-luc, -BIND-WT NRF2, -BIND-Neh4/5mut cells treated with water (control) or 2.5ng/mL dox for 48h, then treated with 2mM NAC for 96h. Data are expressed as a fold relative to parental control (no dox, no NAC), and are derived from two individual replicates performed in triplicate with SD indicated.

The experimental design predominantly relies on a somewhat intricate system where an inducible shRNA downregulates NRF2 while simultaneously inducing the expression of luciferase, WT-NRF2, and various mutants. This complexity may raise questions regarding whether genotype-specific differences stem solely from differing kinetics of expression and downregulation. In Figure 3E, unless I misunderstood something, why the NRF2 ab is unable to recognize the Flag-constructs?

We thank the referee for the opportunity to address these important questions regarding our rescue system. We have encountered significant challenges in exogenously expressing NRF2 under a constitutive promoter in NRF2-dependent cancer cell lines and hence had to rely on an acute inducible system for NRF2 re-expression. Under constitutive expression, when we followed both expression of NRF2 as well as a transcriptionally-linked BFP fluorophore, we observed rapid suppression both within cellular pools as well as when BFP+ cells are enriched by FACS, indicating strong negative selection against increased levels of NRF2. We have tried different strategies and worked for several years to overcome this challenge (trying promoters with varying strength as well as chemical modulators of protein levels such as the DD-tag/Shield1 systems and others). In the end, we identified the dual dox NRF2 knockdown / rescue system we present here as the only successful approach. To our knowledge, we are the first to report a rescue of NRF2 knockdown in a NRF2-dependent cancer cell context (potentially in part to how challenging this was to execute).

To address the concern regarding differences in knockdown across the different cell lines, we have performed qPCR experiments from a dox time course, which shows similar knockdown kinetics and sustained suppression of the endogenous NRF2 transcript. Moreover, we highlight that in Figure 3E and G we present Western blot data showing similar knockdown/re-expression at 24 and 48h, respectively.

qRT-PCR from RNA isolated from A549-shNRF2-BIND-luc, -BIND-WT NRF2, -BIND-Neh4/5mut, or -BIND-ΔNeh1 cells treated with water (control) or 2.5ng/mL dox for 24, 48, or 72h using NFE2L2 primers. Exogenous NRF2 constructs are codon optimized and not detected by the NFE2L2 primer. Data are generated in triplicate with mean +/- SD from one representative experiment shown.

Furthermore, in the revised manuscript, we present rescue of NRF2 knockdown by WT and Neh4/5mut in a distinct cell line, H460shNRF2 (Figure EV4C-E). We feel that collectively this evidence alleviates any concern that the phenotypes we observed are technically artifactual.

Crystal violet staining of H460-shNRF2-BIND-luc, -BIND-WT NRF2, -BIND-Neh4/5mut, or -BIND-ΔNeh1 treated with water (control) or 2.5ng/mL dox for 8 days. Representative data from three independent experiments is shown.

CellTiter-Glo measurements of H460-shNRF2-BIND-luc, -BIND-WT NRF2, -BIND-Neh4/5mut, or -BIND-ΔNeh1 treated with water (control) or 2.5ng/mL dox for 5 days. Data are expressed as a fold of control values, with SD indicated and are derived from three individual experiments.

Specifically addressing the NRF2 antibody question, the antibody does recognize the exogenously expressed NRF2 and can be easily differentiated due to the increased molecular weight caused by the 3X-FLAG tag. In the western included in the original submission we observed unexpectedly low expression of the exogenous NRF2 and this was difficult to detect. We have now repeated this several more times in A549 and have included a new version of this western. In the new Figure 3E exogenous NRF2 levels are similar to endogenous NRF2. We have also included an overlay from the LiCor scan where the FLAG antibody signal was detected in the red channel and the NRF2 antibody signal was detected in the green channel. Figure 3G is a similar experiment in A549 (knockdown of endogenous and re-expression of the exogenous FLAG-WT NRF2 or Neh4,5mut) where we also observe similar levels of the exogenously expressed NRF2 to endogenous NRF2. We have also now included westerns in the 2nd cell line H460 (Figure EV4C). Here we also observe similar levels of exogenous NRF2 to endogenous NRF2.

Why not validate some findings in an NRF2-knockout cell line reconstituted with the inducible system? In our group's experience, A549 cells can be readily engineered to lack NRF2. Establishing such a system would allow for proper quantification of GSH, an aspect somehow unclear. Figure EV6 presents data only in fold versus control, hindering comparisons between genotypes.

In our experience, NRF2 knockout in KEAP1mut cell lines is not suitable for long term culture / experimentation. In the revised manuscript, we present an additional cell line model validating our rescue phenotypes (see above, Figure EV4C-E), and hope that this satisfies the referee's request.

We are providing for the referee raw GSH intensity values underlying our metabolomics data, showing no differences in baseline (-dox) values. The normalized fold changes provided in the manuscript enable a pathway-level analysis of diversely abundant metabolites, facilitating cross-treatment comparisons within the global dataset.

The study would benefit by showing that WT and Neh4 mutants have a similar transactivation activity in an ARE-Luc assays which is routinely employed to address NRF2 activity.

We have frequently used ARE-Luc reporters generated in KEAP1 mutant cell lines. We are aware that the ARE-Luc is an approach that has been used to characterize the activity of various NRF2 mutants or truncations to explain the effect on transcriptional output in KEAP1 WT models, however prior to this request we did not have in-house experience using this system in KEAP1 WT models. We attempted to address this request in two ways. First, we tried transient co-transfection to co-express exogenous NRF2 constructs with an ARE-Luc reporter plasmid in 293s. This failed to show clear NRF2-dependent activation of Luciferase signal over control. Second, we generated 2 KEAP1 WT lung cancer cell lines with ARE-Luc integrated and stably selected. In these lines, we also failed to see strong Luciferase induction upon transfection of exogenously expressed NRF2. Included below are westerns and Luciferase signal from both lines. While we were unable to detect exogenous NRF2 expression in H1299 (likely due to transfection challenges), we did observe expression of exogenous NRF2 in H441. Included as a control in these assays was treatment with a KEAP1 NRF2 PPI inhibitor (from Davies et al. J Med Chem 2016), which resulted in increased NRF2 protein levels and robust Luciferase signal. While the Neh4/5mut looks very similar to WT NRF2 in H441, we are hesitant to include this

data in the final manuscript due to the discrepancy between the strong Luciferase signal with the PPI inhibitor and weak signal for WT NRF2 (and Neh4/5mut) despite similar proteins levels. Also, as we highlighted in our manuscript, regulation of NRF2 targets genes is quite complex and variable across genes and it is hard to know how well this artificial system recapitulates endogenous gene regulation.

H1299 or H441 cells that stably express NanoLuciferase with multiple ARE elements within the promoter region were transfected with constructs constitutively expressing WT NRF2 or Neh4/5mut (or Renilla Luciferase as a control). 48hrs after transfection cells were either lysed for westerns or NanoLuciferase was measured using the Promega Nano-Glo Luciferase Assay. As a control, cells were treated with the KEAP1 NRF2 PPI inhibitor (from Davies et al. J Med Chem 2016) at 1uM for 18hrs.

Exploring the sensitivity of NRF2-deficient, WT-NRF2, and Neh4 mutants to GSH-depleting agents such as BSO and inhibitors of TXNRD1 (e.g., TRi-1) could also provide important insights into the observations reported here.

Thank you for this suggestion, we have performed this experiment and incorporated the data into the revised manuscript (Figure EV7A). The data show that cells rescued by Neh4/5mut exhibit increased sensitivity to BSO, in line with the finding that these cells have diminished antioxidant capacity, consistent with impairments in the GSH pathway.

CellTiter-Glo measurements of A549shNRF2-BIND-luc, -BIND-WT NRF2, -BIND-Neh4/5mut cells treated with water (control) or 2.5ng/mL dox for 48h, then treated with indicated concentrations of BSO for 96h. Data are expressed as a fold relative to parental control (no dox, no BSO), and are derived from two individual replicates performed in triplicate with SD indicated.

The CREBBP/EP300 double KO cells are not satisfactorily characterized. Do they also show decreased GSH levels? Do they show loss of expression of group1 NRF2 target genes?

We thank the referee for this comment, and apologize that it was unclear in our original submission that the dual *CREBBP/EP300* KO data were from a publicly available source (Ito et al 2021). These data were invoked to further validate growth phenotypes stemming from our rescue system and experiments with A-485.

We have chosen to address this comment by performing further experiments with A-485, which permits specific and simultaneous inhibition of both paralogs. We performed both metabolomics and RNA-seq in A549 cells. Indeed, we observed decreased levels of GSH in addition to pathway level defects with A-485 treatment (24h, 1uM) in A549 cells, consistent with increases in ROS (Figure 5G). In RNA-seq experiments, we observed greater suppression of group I vs group II genes by RNA-seq, in agreement with our findings that these genes are more enhancer-dependent. These data are now included in Figure EV7B&C.

(left) RNA-seq values of group I vs group II (Figure 1B) genes from A549 and H460 cells treated with 1μM A-485 for 6h. Values show limma-voom log₂ fold changes between A-485 treatment vs DMSO. Data are derived from three individual replicates. P-values derived from a Wilcoxon rank sum test, followed by adjustment by the Benjamini-Hochberg procedure to correct for multiple testing. (right) Heatmap of GSH pathway metabolites intensity normalized by row as a z score from A549shNRF2-BIND-luc cells treated with 1μM A-485 for 24h. Each column represents an individual replicate.

Referee #3:

In their study "NRF2 drives non-small cell lung cancer growth independently of CBP/p300-enhanced glutathione synthesis" the authors use several omic-based screens, genomic and bioinformatic analyses to evaluate the role of NRF-2 in lung cancer cells.

This data-reach study addresses an important aspect of cancer cell biology i.e. the involvement of redox regulation. As such, I am convinced that the datasets provided here might be of use for researchers working in the field.

We thank Referee #3 for reviewing our work, and appreciate the value they see for researchers in the NRF2 and redox fields.

Comments and suggestions:

- A more detailed yet understandable explanation of how the 59 NRF2-controlled target genes were selected would be helpful.

Thank you for this comment, we have modified the text and figure legends to provide a more detailed and understandable explanation.

- All immunoblots need quantification and statistical evaluation, i.e., they should be repeated multiple times.

We have repeated all Western blots presented multiple times and present them as source data, in accordance with EMBO Press guidelines.

- The cell biology-based evaluation of the omic/genomic and bioinformatic results should be more robust to support the study's claims (see title and abstract).

We interpret this comment as a suggestion to tone down claims made in the title and abstract, similar to what referee #1 suggested. We have made text edits in the title (i.e., replacing “drives” with “supports”) and abstract (specifying work is derived from cell line models) to moderate some of our claims. In addition, we have performed additional cell biology experiments (i.e., rescue in a distinct cell line (Figure EV4C-E), BSO treatments (Figure EV7A), A-485 metabolomic and RNAseq experiments (EV7B)). We hope that collectively these changes satisfy the referee’s request.

- Using only one technique to evaluate cell viability is insufficient. Cell titer glow is a detection system that depends on cell metabolic state, which might be problematic in a situation where cell metabolism is affected (following NRF2 down-regulation for example)

We agree with the reviewer that orthogonal assays will strengthen our interpretations, in particular using readouts that directly measure cellular abundance and do not rely on a metabolic readout. We have provided orthogonal assays for both A-485 treatment (Cell-titer glo in Fig 4C & clonogenic for a subset of the lung lines in Fig EV5B) and NRF2 knockdown/re-expression (Incucyte in Fig 4A & clonogenic in 4B, EV4D).

- Additional measurements of cancer cell biology, such as migration and invasion assays, would help in understanding the contribution of the findings to cancer cell biology.

We appreciate the intention of further validating our findings using additional methods of cancer cell biology. However, the focus of our paper has been on both the mechanistic dissection of NRF2 transcriptional regulation and its impact on the viability of NRF2-dependent cells. We feel that assays interrogating migration and metastasis phenomena are out of scope of the current manuscript. Nevertheless, we have performed scratch assays in A549shNRF2-BIND cells, and observed that WT and Neh4/5mut display enhanced wound healing capacity relative to NRF2 knockdown/rescue by Luciferase or Δ Neh1.

A549shNRF2-BIND-luc, WT, Neh4/5mut, Δ Neh1 cells were treated +/- 2.5ng/mL dox for 24h, followed by scratching using a cell scraper. Cells were washed 3X with PBS, media (+/- 2.5ng/mL) was replaced, and cells were subjected to Incucyte imaging for 7 days. Scratch area immediately following the scratch and at day 7 was quantified using the Wound Healing Size Tool (Suarez-Arnedo et al, 2020) in ImageJ/Fiji. Percent scratch area healed represents the percent of the initial scratch area covered with cells at day 7, with three individual replicates presented.

- Detecting GSH and H₂O₂ using commercial kits known to be prone to artifacts is risky and may affect the conclusions. More advanced techniques, such as EPR or genetically encoded sensors, should be considered for detecting H₂O₂ and GSH/GSSG.

We appreciate this concern, and agree that more recent techniques would be a sophisticated way to demonstrate effects on antioxidant capacity. However, we would like to underscore that commercial GSH kits were used mainly to validate findings stemming from unbiased metabolomic mass spectrometry experiments. We observed pathway-level effects on GSH (i.e., GSH-related dipeptides, Figure 5) strengthening confidence in our data from both metabolomics and from commercially available kits.

As for ROS measurements, we have performed experiments using the more classical ROS sensor, DCF, in A549 cells under various perturbations. We include this data for the referee below. We generally find that DCF and the H₂O₂ assays are well correlated. Unfortunately, because of the use of fluorescent markers in our cell line engineering experiments, DCF is not suitable for ROS measurement in our rescue system.

A549 cells were treated with A-485 (1 μ M), siNRF2 (25nM, transfected with RNAiMax according to manufacturer's protocol), or BSO (250 μ M) for 24h, followed by staining with 5 μ M DCF for 30min and analysis by flow cytometry using standard procedures. Data represent three independent experiments, with SD indicated.

- A second cancer cell entity should be used for the most relevant experiments. For example, melanoma cells seem to be highly reliable on redox regulation.

We agree that a second cancer cell model would strengthen our claims, and we have provided an additional NSCLC cancer cell line (H460) to validate many of our findings (Figure EV4C-E). The current manuscript focuses on NSCLC, the main indication where NRF2 is chronically activated in cancer, and we feel that melanoma cells would be out of scope of the current work.

- Using inhibitors of GSH synthesis, such as BSO or Auranofin, as well as H₂O₂ scavengers, could help verify the authors' statements regarding the role of these two metabolites in cancer cell biology.

Thank you for this suggestion, we have performed this experiment and incorporated the data into the revised manuscript (Figure EV7A). The data show that cells rescued by Neh4/5mut exhibit increased sensitivity to BSO, in line with the finding that these cells have impaired GSH pathway / lower antioxidant capacity.

Referee #4:

The manuscript by Conrad et al. provides a very innovative and elegant piece of work in which the authors robustly demonstrate that Nrf2 binding to its co-activator CBP/p300 is required for transcription of a set of genes involved in GSH-de novo synthesis but not in cancer cell proliferation. The study is extremely well done. Only a few concerns need to be addressed.

We thank the referee for reviewing our manuscript and for this kind evaluation of our work.

The main one is: "How broad and comprehensive is the message the authors want to convey?" Looking at the results, it seems that the only Nrf2 ablation (with any external oxidative insult) leads to a significant decrease of GSH (and glutamate-containing dipeptides) and an increase in H₂O₂ levels. This, however, has no effect on cell proliferation, as proven by the authors. At the same time, conditions of Nrf2 knockdown lead to a decrease of one glycolytic metabolite (i.e., F1,6-BP) and an increase of sedoheptulose-7-P, which is a marker of PPP. By comprehensively looking at these results, I've gotten the impression that Nrf2-downregulation leads to general metabolic reprogramming. Specifically, it seems that Nrf2-KO cells redirect glucose to the PPP for the generation of NADPH and provide reducing equivalents to the antioxidant system, which is not "transcriptionally sustained" by Nrf2. This makes glutamine the principal metabolic source for obtaining ATP from the mitochondria (hence the increase of H₂O₂). The decrease in nucleotide levels further corroborates the idea that glucose undergoes oxidative decarboxylation primarily to produce NADPH via the oxidative branch of the PPP without being converted to ribose or giving rise to IMP or CMP.

We thank the referee for this useful comment and intriguing hypothesis of metabolic reprogramming upon depletion of NRF2. We agree that disentangling the role of NRF2 is this process, in particular in the context of PPP is quite important. It is reported that the oxidative branch (that generates NADPH for antioxidant and reductive biosynthesis) and non-oxidative branch that reversibly shuttles ribose phosphate and glycolytic intermediates depending on demand) can operate independently or together (reviewed in Stincone et al. 2014). The metabolic phenotype of F16BP reduction and S7P accumulation (as noted by the reviewer) upon NRF2 knockdown may be associated with increased demand in ribose for nucleotide synthesis in line with increased IMP (purine biosynthetic intermediate). This activity can be independent of the oxidative branch of PPP and NADPH production.

To experimentally determine metabolic flux between glycolysis and oxidative PPP (oxPPP) upon NRF2 depletion, we have performed isotope tracing experiments with [1,2-¹³C]glucose in the context of NRF2 knockdown (A549 shNRF2-luc +/- dox) and NRF2 activation (H441 treated with a KEAP1 NRF2 PPI inhibitor from Davies et al. J Med Chem 2016). We observed that oxPPP flux relative to glycolysis is decreased under NRF2 knockdown (Figure EV6C), arguing against the proposed model of metabolic reprogramming. Furthermore, we observe the opposite trend of oxPPP flux under NRF2 activation (Figure EV6D). These data are in agreement with the prevailing model of NRF2 driving oxPPP / NADPH levels (Mitsuishi 2012).

Our interpretation is that the reviewer has proposed this metabolic programming phenotype in response to increased NADP levels upon NRF2 knockdown (luc condition) in our global metabolomics data set. This phenotype has indeed puzzled us as well, as we would expect NRF2 knockdown to decrease the NADP pool and/or NADPH levels. Accurately quantifying NADP⁺/NADPH in global metabolomics is challenging, not only due to interconversion, but also due to the documented instability of these metabolites to particular solvents and increased temperature during metabolite extraction, which are inherent to LC/MS required for the Mass Spec analysis (Lu et al 2018). We have since undertaken a targeted NADP⁺/NADPH assay and

observed decreases in both NADP⁺ and NADPH in response to NRF2 knockdown. These data, as well as clarifying statements regarding the NADP phenotype in global metabolomics dataset, are now included in the manuscript (Figure EV6A&B). We hope that these follow up studies address the referee's concerns.

An explanation of all this rewiring is that the effects the authors are seeing might be due to clonal selection. How many clones did they screen? The authors should at least perform other analyses in a cell system known to be not reliant on glutamine or (alternatively) perform additional experiments with different A-549 clones grown in media that are not supplemented with glutamine, glucose, or inhibited to use lipids to fuel mitochondrial metabolism. This consideration is even more relevant, given the altered lipid composition the authors observe in this condition.

In addition to the further studies validating metabolic phenotypes upon NRF2 knockdown that we describe above, we would like to clarify that the system represents a polyclonal pool. Although the A549 shNRF2 cell line in which we introduced NRF2 expression constructs started from a single cell clone, the rescue system represents many, many doublings of this original clone. We therefore argue that the phenotypes presented are less likely to be impacted by clonal selection. Moreover, we provide an additional cell line validating our rescue phenotypes in the revised manuscript (Figure EV4C-E). We feel that these points sufficiently address any concerns regarding clonal artifacts.

Investigating whether these cells (more clones) are more vulnerable to H₂O₂-mediated insults or whether they have evolved to withstand oxidative stress is another important topic that deserves to be considered.

Thank you for this suggestion, we have performed this experiment and incorporated the data into the revised manuscript (Figure EV7A). The data show that cells rescued by Neh4/5mut exhibit increased sensitivity to BSO, in line with the finding that these cells have impaired antioxidant capacity.

CellTiter-Glo measurements of A549shNRF2-BIND-luc, -BIND-WT NRF2, -BIND-Neh4/5mut cells treated with water (control) or 2.5ng/mL dox for 48h, then treated with indicated concentrations of BSO for 96h. Data are expressed as a fold relative to parental control (no dox, no BSO), and are derived from two individual replicates performed in triplicate with SD indicated.

Minor points:

- The authors should add more details in the figure legends. They are too short and provide no information or clues to understand the figures.

Thank you for this comment, we have made substantial revisions to the figure legends.

- The authors should provide molecular weights on the western blots. Especially for Nrf2 (whose molecular weight is always debated), this information is fairly relevant.

Thank you, we have added molecular weight markers in the revised manuscript.

- Besides GSH values, the authors should also provide the extent of GSSG. This value is important to understand whether the increase in GSH observed upon re-expression of WT Nrf2 is due (mainly or uniquely) due to an increase in the rate of de novo synthesis or reduction. In the light of what discussed above, defining this issue is very relevant.

We agree with this point and have performed this experiment which is now included as Figure EV6F. The data show that there is no change in the GSH/GSSG ratio, indicating that changes observed are due to *de novo* synthesis rates and not reduction.

GSH/GSSG ratios in A549shNRF2-BIND-luc, -BIND-WT NRF2, -BIND-Neh4/5mut cells treated with water (control) or 2.5ng/mL dox for 48h. rescued with luciferase control, WT or Neh4/5mut NRF2. Data represent three independent experiments with SD indicated.

References

Davies, T.G., Wixted, W.E., Coyle, J.E., Griffiths-Jones, C., Hearn, K., McMenamin, R., Norton, D., Rich, S.J., Richardson, C., Saxty, G., *et al.* (2016). Monoacidic Inhibitors of the Kelch-like ECH-Associated Protein 1: Nuclear Factor Erythroid 2-Related Factor 2 (KEAP1:NRF2) Protein-Protein Interaction with High Cell Potency Identified by Fragment-Based Discovery. *J Med Chem* 59, 3991-4006.

Lu, W., Wang, L., Chen, L., Hui, S., and Rabinowitz, J.D. (2018). Extraction and Quantitation of Nicotinamide Adenine Dinucleotide Redox Cofactors. *Antioxid Redox Signal* 28, 167-179.

Mitsuishi, Y., Taguchi, K., Kawatani, Y., Shibata, T., Nukiwa, T., Aburatani, H., Yamamoto, M., and Motohashi, H. (2012). Nrf2 Redirects Glucose and Glutamine into Anabolic Pathways in Metabolic Reprogramming. *Cancer Cell* 22, 66-79.

Stincone, A., Prigione, A., Cramer, T., Wamelink, M.M., Campbell, K., Cheung, E., Olin-Sandoval, V., Gruning, N.M., Kruger, A., Tauqeer Alam, M., *et al.* (2015). The return of metabolism: biochemistry and physiology of the pentose phosphate pathway. *Biol Rev Camb Philos Soc* 90, 927-963.

Suarez-Arnedo, A., Torres Figueroa, F., Clavijo, C., Arbelaez, P., Cruz, J.C., and Munoz-Camargo, C. (2020). An image J plugin for the high throughput image analysis of in vitro scratch wound healing assays. *Plos One* 15, e0232565.

Dear Dr. Foster

Thank you for the submission of your revised manuscript to EMBO reports. We have now received the full set of referee reports that is copied below.

As you will see, the referees find that the study has been significantly strengthened during the revision. Referee #3 remains concerned that the conclusions on cancer biology are largely based on cell proliferation and viability. Upon further discussion of this point, the referee re-emphasized the importance of evaluating not just proliferation but also invasion. The referee noted: "[...] I recommend measuring invasion and migration based on my own experience and published findings, which highlight the central role of redox signaling in metastatic spread. Since low proliferative, senescence-like behavior is frequently associated with increased metastatic burden, I believe that these parameters should be assessed in all cancer biology studies. Failing to include such analyses may lead a non-expert reader to misinterpret the true effects of certain treatments or genetic manipulations. For example, existing literature suggest that antioxidant treatments protect metastatic cells and promote tumor aggressiveness (e.g., via ferroptosis). However, this conclusion is only valid when evaluating not just proliferation but also invasion. I, hence, believe these experiments are essential. While in vivo experiments would be ideal they are also time consuming. However, in vitro migration and invasion assays are more feasible and, in my opinion, represent a reasonable requirement."

In order to strengthen your conclusions, please address this remaining requests by performing in vitro migration and invasion assays.

From the editorial side, there are also a few things that we need before we can proceed with the official acceptance of your study.

- Please provide up to 5 keywords
- Please update the 'Conflict of interest' paragraph to our new 'Disclosure and competing interests statement'. For more information see <https://www.embopress.org/page/journal/14693178/authorguide#conflictsofinterest>
- Please correct the following name discrepancy: Weng Ruh Wong in the manuscript vs. Weng Wong in the online manuscript tracking system.
- Regarding the Author Contributions, we now use CRediT to specify the contributions of each author in the journal submission system. Therefore, please remove the Author Contributions from the manuscript file and make sure that the author contributions in our online manuscript tracking system are correct and up-to-date. The information you specified in the system will be automatically retrieved and typeset into the article. You can enter additional information in the free text box provided, if you wish.
- Please provide an ORCID ID for co-corresponding author Dr. Sterne-Weiler. Please find instructions on how to link your ORCID ID to your account in our manuscript tracking system in our Author guidelines (<https://www.embopress.org/page/journal/14693178/authorguide#authorshipguidelines>)
- The manuscript sections should be in the following order: Title page - Abstract & Keywords - Introduction - Results - Discussion - Methods - Data Availability - Acknowledgments - Disclosure Statement & Competing Interests - References - Figure Legends - (Main Tables with legends if applicable) - Expanded View Figure Legends.
- Please provide information on funding in the manuscript's Acknowledgment section and in the online manuscript tracking system.
- Please provide all main and EV figures as separate production quality Figure files.
- On page 6 you refer to 'Extended Data 1', but I could not find the data you refer to. Could you please clarify and update this callout? The genome browser snapshots could be shown in an Appendix PDF.
- You have currently 8 EV tables, however, the following tables are more complex and should be uploaded and renamed to datasets: Table EV2, Table EV4, Table EV5, Table EV6, Table EV7, Table EV8 (should be Dataset EV1-EV5); EV table legends need to be removed from the manuscript and provided in each Excel file (for Datasets, each legend can be provided as a separate sheet/tab in the Excel file).
- Please add a callout for Table EV8.
- Please provide a Reagents and Tools Table listing key reagents, experimental models, software and relevant equipment and including their sources and relevant identifiers.

You can find and download the Reagents and Tools Table template (.docx) in our author guidelines:

- Citations of preprints: please add the prefix 'preprint:' to in-text citations and [PREPRINT] at the end of the reference in the reference list.

- Please fill column D in the author checklist by choosing the appropriate answer from the pull-down menu.

Figure legends:

- Figure EV1d: please complete the last sentence in the legend for this panel. If the data are derived from one experiment, please show the individual datapoints instead of the mean and SEM. What is the mean based on? Technical replicates? Please specify this.

- Figure 4a: the legend states that you show representative data from two independent experiments. Can you please define 'n' for the data shown? Are mean and SEM derived from technical replicates?

- For all graphs showing quantification: please show the individual data points in addition to the mean and error bars.

- Please note that the figure panel 5d is not labelled in the figure, however the corresponding legend is labelled as 5d in the manuscript. This needs to be rectified.

- Please provide descriptive figure titles for figures EV 1-7 (legends).

- Please provide the exact p values in the legends of figures 1d; 2e; 5f-g.

- Please note that scale bar and its definition are missing for figures 4b; EV 4d.

- Please note that axis labels are not defined for figure 3b.

- Please note that axis gaps are not labeled appropriately in figure 3d.

- Finally, EMBO Reports papers are accompanied online by

A) a short (1-2 sentences) summary of the findings and their significance,

B) 2-3 bullet points highlighting key results and

C) a schematic summary figure that provides a sketch of the major findings (not a data image).

Please provide the summary figure as a separate file in PNG or JPG format at a size of 550x300-600 pixels (width x height).

Please note that the size is rather small and that text needs to be readable at the final size. Please send us this information along with the revised manuscript.

With kind regards,

Martina Rembold, PhD

Senior Editor

EMBO reports

=====

Referee #1:

The authors have satisfactorily addressed my comments with new experimental data, and when not, they were able to provide a well-thought-through justification.

Referee #3:

The authors conducted several additional experiments, resulting in an improved manuscript. However, I disagree with their statement regarding cell migration and invasion and their refusal to experimentally evaluate these critical cellular traits, particularly when focusing on cancer cell biology. While viability (proliferation) is indeed important, it should not be assessed in isolation without understanding the impact on migration and invasion, etc. It is well-established that in several cancer types, a decreased proliferation rate can lead to enhanced migration and metastatic spread. Additionally, measuring H₂O₂ using DCF is, in my opinion, problematic due to the numerous artifacts associated with this method. These experiments should be repeated using more reliable experimental tools.

Referee #4:

The authors addressed all my concerns and convinced me about my major criticism

Referee #1:

The authors have satisfactorily addressed my comments with new experimental data, and when not, they were able to provide a well-thought-through justification.

We thank the referee for their time and effort in reviewing our manuscript.

Referee #3:

The authors conducted several additional experiments, resulting in an improved manuscript. However, I disagree with their statement regarding cell migration and invasion and their refusal to experimentally evaluate these critical cellular traits, particularly when focusing on cancer cell biology. While viability (proliferation) is indeed important, it should not be assessed in isolation without understanding the impact on migration and invasion, etc. It is well-established that in several cancer types, a decreased proliferation rate can lead to enhanced migration and metastatic spread.

Additionally, measuring H₂O₂ using DCF is, in my opinion, problematic due to the numerous artifacts associated with this method. These experiments should be repeated using more reliable experimental tools.

We thank the referee for this feedback and for sharing their expertise in the context of our manuscript. We have now included the wound healing assay, originally only included for referees in the rebuttal, in the manuscript as Figure EV4F.

We have supplemented the text with the following statement in the results section:

“Both WT and Neh4/5mut NRF2 rescue cells exhibited comparable outgrowth in a scratch wound healing assay (Figure EV4F), indicating that CBP/p300 loss-of-function does not impact migratory capacity of cells on tissue culture plates.”

And the following in the discussion section:

“Also of importance, and beyond the scope of the current 2D cell culture assays (Figure 4, Figure EV4), is understanding the impact of CBP/p300-enhanced NRF2 redox signaling on cancer cell invasion/migration within the framework of metastatic disease progression, ultimately in in vivo systems”

We anticipate that these additions will augment the cell biological evaluation of our claims while clearly specifying that the scope of the current work is 2D proliferation. We recognize that more sophisticated *in vivo* experiments would be better suited to address any effects of CBP/p300 loss-of-function on NRF2-dependent cancer cell migration/invasion. However, these studies are not only time- and resource-intensive, but we consider them supplementary to the primary

objectives of the current mechanistic study of the NRF2/CBP/p300 interaction on proliferation using *in vitro* models.

We acknowledge the inherent limitations associated with using DCF as a ROS sensor and conducted these experiments to corroborate data obtained from more specific H₂O₂ assays. The use of fluorophores in our rescue system necessitated a careful selection of ROS detection methods. The consistency between luminescent- and DCF-based methods, together with the observed pathway-level disruptions in the GSH system (Figure 5, Figures EV6, 7), reinforces our confidence in the validity of the reported findings.

Referee #4:

The authors addressed all my concerns and convinced me about my major criticism

We appreciate the constructive feedback and thoughtful review of our work.

**The authors have addressed all minor editorial requests.

Dr. Scott Foster
Genentech
Discovery Oncology
1 DNA Way
South San Francisco, CA 94080
United States

Dear Dr. Foster,

I am very pleased to accept your manuscript for publication in the next available issue of EMBO reports. Thank you for your contribution to our journal.

Yours sincerely,
